# The meiotic LINC complex component KASH5 is an activating adaptor for cytoplasmic dynein

Kirsten E.L. Garner[1]*, Anna Salter[1,2]*, Clinton K. Lau[3], Manickam Gurusaran[4], Cécile M. Villemant[1], Elizabeth P. Granger[1], Gavin McNee[1], Philip G. Woodman[1], Owen R. Davies[4], Brian E. Burke[2]*, and Victoria J. Allan[1,2]*

Cytoplasmic dynein-driven movement of chromosomes during prophase I of mammalian meiosis is essential for synapsis and genetic exchange. Dynein connects to chromosome telomeres via KASH5 and SUN1 or SUN2, which together span the nuclear envelope. Here, we show that KASH5 promotes dynein motility in vitro, and cytosolic KASH5 inhibits dynein's interphase functions. KASH5 interacts with a dynein light intermediate chain (DYNC1LI1 or DYNC1LI2) via a conserved helix in the LIC C-terminal, and this region is also needed for dynein's recruitment to other cellular membranes. KASH5's N-terminal EF-hands are essential as the interaction with dynein is disrupted by mutation of key calcium-binding residues, although it is not regulated by cellular calcium levels. Dynein can be recruited to KASH5 at the nuclear envelope independently of dynactin, while LIS1 is essential for dynactin incorporation into the KASH5–dynein complex. Altogether, we show that the transmembrane protein KASH5 is an activating adaptor for dynein and shed light on the hierarchy of assembly of KASH5–dynein–dynactin complexes.

## Introduction

To conceive healthy offspring, a paternal sperm and maternal egg must be created, which requires the specialized form of cell division, meiosis, in which maternal and paternal homologs pair in prophase I to allow the swap of genetic material by synapsis, generating genetically distinct haploid daughter cells. This is facilitated by chromosomes attaching to the nuclear envelope (NE), usually via their telomeres (Kim et al., 2022; Rubin et al., 2020). In vertebrates and many other organisms, the chromosomes move along the inner nuclear membrane (INM) to transiently cluster in a "meiotic bouquet" (Fig. S1 A) that brings homologs into close proximity and promotes the formation of the synaptonemal complex in zygotene, enhancing synapsis and recombination (Burke, 2018; Kim et al., 2022; Rubin et al., 2020). In many organisms, these dynamic chromosome movements require force generated by the microtubule motor cytoplasmic dynein-1 (dynein, hereafter) either acting directly on the chromosomes (Horn et al., 2013; Lee et al., 2015; Morimoto et al., 2012; Rog and Dernburg, 2015; Sato et al., 2009; Wynne et al., 2012) or indirectly by driving nuclear movement (Burke, 2018; Lee et al., 2015; Rubin et al., 2020). Crucially, the force must be

transmitted from the cytoplasm to the chromosomes on the other side of the NE. This is achieved by linkers of nucleoplasm and cytoplasm (LINC) complexes, which span the NE to physically connect the cytoskeleton and nucleus (Burke, 2018; Sato et al., 2009; Spindler et al., 2019; Fig. S1 A).

LINC complexes consist of SUN (Sad1, Unc-84) domain proteins that span the INM, binding nuclear lamins and chromatin-associated proteins in the nucleoplasm and interacting with KASH (Klarsicht, ANC-1, Syne Homology) domain proteins. KASH proteins have large cytosolic domains that bind cytoskeleton-associated proteins and are anchored in the outer nuclear membrane (ONM) by a C-terminal transmembrane domain (Burke, 2018; Kim et al., 2022; Rubin et al., 2020). Inside the NE lumen, the short C terminal KASH-domain sequence associates with SUN proteins, restricting KASH proteins to the nuclear membrane (Hao and Starr, 2019; Morimoto et al., 2012; Starr and Han, 2002). The meiotic LINC complex can contain SUN1 and SUN2, which have both overlapping and distinct roles in meiosis (Ding et al., 2007; Lei et al., 2009; Link et al., 2014; Schmitt et al., 2007). Although $Sun1^{-/-}$ mice are sterile (Chi et al., 2009), SUN2

---

[1]School of Biological Sciences, Faculty of Biology, Medicine and Health, University of Manchester, Manchester, UK; [2]A*STAR Institute of Medical Biology, Singapore, Singapore; [3]MRC Laboratory of Molecular Biology, Francis Crick Avenue, Cambridge Biomedical Campus, Cambridge, UK; [4]Wellcome Centre for Cell Biology, Institute of Cell Biology, University of Edinburgh, Edinburgh, UK.

*K.E.L. Garner, A. Salter, B.E. Burke, and V.J. Allan contributed equally to this paper. Correspondence to Victoria J. Allan: viki.allan@manchester.ac.uk

Brian Burke's current affiliation is A*STAR Skin Research Labs (A*SRL), Biopolis, Singapore, Singapore. Elizabeth P. Granger's current affiliation is University of Central Lancashire, Preston, UK. Clinton K. Lau's current affiliation is Department of Biochemistry, University of Oxford, Oxford, UK. Gavin McNee's current affiliation is University of Birmingham, Edgbaston, Birmingham, UK. Anna Salter's current affiliation is Immunocore Ltd, Milton Park, UK. Cécile Madeleine Villemant's current affiliation is The Automation Partnership, (Cambridge) Limited, Hertfordshire, UK. A preprint of this manuscript was posted in bioRxiv on April 13, 2022.

can recruit telomeres to the NM and drive bouquet formation (Link et al., 2014; Schmitt et al., 2007) and allow some prophase chromosome movement (Lee et al., 2015). SUN1 and 2 bind to a meiotic telomere complex (Dunce et al., 2018; Shibuya et al., 2014) and to KASH5 in vertebrates or ZYG-12 in *Caenorhabditis elegans*, which recruit dynein to the ONM (Fig. S1 A; Horn et al., 2013; Lee et al., 2015; Morimoto et al., 2012; Rog and Dernburg, 2015; Sato et al., 2009; Wynne et al., 2012). KASH5 is essential for mammalian meiosis as *Kash5*−/− mice are completely sterile due to impaired synapsis, accumulation of double-stranded DNA breaks, and resulting meiotic arrest (Horn et al., 2013). Chromosome movement and nuclear rotation in prophase I are also lost (Lee et al., 2015). Human mutations that cause mistargeting of KASH5 to mitochondria lead to male sterility (Bentebbal et al., 2021). KASH5 shares sequence homology with the N-terminal region of the protein encoded by the zebrafish gene *futile cycle*, which is required for pronuclear migration (Dekens et al., 2003; Lindeman and Pelegri, 2012), another dynein-dependent function that drives the female pronucleus toward the male pronucleus along microtubules nucleated from the male centrosome (Gönczy et al., 1999; Payne et al., 2003; Reinsch and Karsenti, 1997; Robinson et al., 1999). ZYG-12 performs the same role in *C. elegans* (Malone et al., 2003), but the KASH protein responsible for pronuclear migration in mammals is not yet known.

Dynein transports a wide range of cargo to the minus end of microtubules (Reck-Peterson et al., 2018). It is a large 1.6 MDa holoenzyme (Fig. S1 B) comprised of two heavy chains (DHC: DYNC1H1) containing the globular motor domains, which are the sites of ATP hydrolysis. Its cargo binding tail domain contains the intermediate chains (ICs: DYNC1I1 and 2) that bind directly to the DHCs; three light chains that bind to the ICs; and two light intermediate chains (LICs: DYNC1LI1 and 2; Pfister et al., 2005; Reck-Peterson et al., 2018). The LICs bind to DHC via their N terminal GTPase-like domain, which is highly conserved between LICs 1 and 2 (Schroeder et al., 2014). The unstructured carboxy terminus protrudes from the motor complex (Celestino et al., 2019; Lee et al., 2018; Schroeder et al., 2014) and is less homologous between LIC1 and 2 apart from two regions of predicted alpha-helix that link dynein to cargo adaptors (Celestino et al., 2022; Celestino et al., 2019; Kumari et al., 2021b; Lee et al., 2020; Lee et al., 2018).

Dynein requires the multicomponent dynactin complex (Fig. S1 B) for function (Feng et al., 2020; Gill et al., 1991; King et al., 2003; McKenney et al., 2014; Schlager et al., 2014a; Schroer and Sheetz, 1991), although dynein and dynactin alone interact weakly (Baumbach et al., 2017; Chowdhury et al., 2015; Jha et al., 2017; Splinter et al., 2012; Urnavicius et al., 2015). An "activating adaptor" (Fig. S1 B) is needed for optimum motility and force generation in vitro (Belyy et al., 2016; McKenney et al., 2014; Schlager et al., 2014a; for reviews see Canty and Yildiz, 2020; Olenick and Holzbaur, 2019; Reck-Peterson et al., 2018). They strengthen the interaction between dynein and dynactin (Schroeder and Vale, 2016; Splinter et al., 2012) by forming extensive interactions with both components to generate a tripartite dynein–dynactin–adaptor (DDA) complex or complexes with two dyneins per dynactin and an adaptor (D₂DA; Chowdhury et al., 2015; Grotjahn et al., 2018; Lau et al., 2021; Lee et al., 2020; Urnavicius et al., 2018; Urnavicius et al., 2015) or two dyneins, one dynactin, and two adaptors (Chaaban and Carter, 2022). Adaptor binding releases the dynein motor domains from the autoinhibited Phi conformation (Torisawa et al., 2014; Zhang et al., 2017), helping to align them for microtubule interaction (Chowdhury et al., 2015; Zhang et al., 2017). Finally, adaptors recruit dynein and dynactin to cargo (e.g., Horgan et al., 2010b; Schroeder and Vale, 2016; Splinter et al., 2012; Wang et al., 2019).

The assembly and function of dynein adaptor complexes are promoted by Lissencephaly-1 (LIS1). LIS1 is needed for many dynein functions (Markus et al., 2020) and is mutated in the neurodevelopmental disorder, Type-1 lissencephaly (Reiner et al., 1993). LIS1 binding to the dynein motor domain opens the Phi complex (Elshenawy et al., 2020; Gillies et al., 2022; Htet et al., 2020; Marzo et al., 2020; Qiu et al., 2019) and enhances the formation of DDA complexes (Baumbach et al., 2017; Dix et al., 2013), increasing the frequency, velocity, and duration of dynein movement (Baumbach et al., 2017; Dix et al., 2013; Fenton et al., 2021; Gutierrez et al., 2017). LIS1 also increases the proportion of the two dynein D₂DA complexes (Elshenawy et al., 2020; Htet et al., 2020), which generate more force, move faster, and are more processive than DDA complexes (Elshenawy et al., 2019; Sladewski et al., 2018; Urnavicius et al., 2018). Furthermore, LIS1 contributes to the recruitment of dynein, dynactin, and/or adaptors to a wide range of cellular cargoes in interphase and mitosis (Cockell et al., 2004; Dix et al., 2013; Dzhindzhev et al., 2005; Lam et al., 2010; Siller et al., 2005; Sitaram et al., 2012; Splinter et al., 2012; Wang et al., 2013).

Activating adaptor proteins share little sequence homology, but generally contain a long coiled-coil domain, and a site for cargo binding (Lee et al., 2020; McKenney et al., 2014; Redwine et al., 2017; Schlager et al., 2014a; Schlager et al., 2014b; Urnavicius et al., 2015). They bind the C-terminal domain of LICs via at least three distinct types of sequence motif—the CC1 box, the Hook domain, or EF hands (Celestino et al., 2019; Gama et al., 2017; Lee et al., 2020; Lee et al., 2018; Schroeder and Vale, 2016)—and the motility of all three types of complexes is promoted by LIS1 (Htet et al., 2020). Other adaptors may not activate dynein but still link the motor to cargos (Olenick and Holzbaur, 2019; Reck-Peterson et al., 2018). For example, Rab7-interacting lysosomal protein (RILP) recruits dynein and dynactin to Rab7-positive late endosomes/lysosomes (Johansson et al., 2007; Jordens et al., 2001; Scherer et al., 2014; Tan et al., 2011) via the binding of its RILP homology domains to helix 1 of LIC (Celestino et al., 2022), but RILP lacks a long coiled coil. The KASH proteins KASH5 and ZYG-12 are good activating adaptor candidates because they bind dynein and have long coiled coils, with KASH5 having an N-terminal EF hand domain (Horn et al., 2013; Morimoto et al., 2012), whereas ZYG-12 has a Hook domain (Malone et al., 2003). However, unlike other activating adaptors identified so far, they are transmembrane proteins.

Mammalian dynein contains either two LIC1 subunits or two LIC2 subunits (Tynan et al., 2000a), providing the potential for differential interactions with adaptors. However, both LICs bind to the adaptors Rab11-FIP3 (Celestino et al., 2019; Horgan et al.,

2010a; Horgan et al., 2010b), RILP (Celestino et al., 2019; Scherer et al., 2014), BICD2, spindly, Hook3, ninein, and TRAK1 (Celestino et al., 2019) via the highly conserved helix 1 in the LIC C-terminus. This shared binding ability suggests that LICs may act redundantly, as has been reported in the endocytic pathway (Horgan et al., 2010a; Horgan et al., 2010b; Tan et al., 2011). However, other endocytic functions may be isoform-specific (Bielli et al., 2001; Hunt et al., 2013; Palmer et al., 2009). Likewise, although LIC1 and 2 act redundantly for some mitotic functions (Jones et al., 2014; Raaijmakers et al., 2013), isoform-specific functions and localizations have been reported in mitosis (Horgan et al., 2011; Mahale et al., 2016a; Mahale et al., 2016b; Palmer et al., 2009; Raaijmakers et al., 2013; Sharma et al., 2020; Sivaram et al., 2009), at the centrosome (Tynan et al., 2000b), during neuronal nuclear migration (Goncalves et al., 2019), and cell migration (Even et al., 2019; Schmoranzer et al., 2009). In addition, mitotic phosphorylation of the LIC1 C-terminal domain may offer temporal control of adaptor selection (Kumari et al., 2021a). The degree of specific and shared functions for LICs is an important issue that is not fully resolved.

How the dynein motor interacts with KASH5 to drive the dynamic chromosome movements essential for mammalian meiosis has been poorly understood. Here, we reveal KASH5 to be a dynein-activating adaptor that interacts with the LIC helix 1 region, in agreement with recent work (Agrawal et al., 2022). We show that this region in the LIC is also key for dynein's function at the Golgi apparatus and throughout the endocytic pathway, with LICs 1 and 2 acting redundantly. Analysis of the hierarchy of adaptor complex assembly reveals that dynein can be recruited to KASH5 independently of dynactin and that LIS1 is essential for full complex assembly. The interaction between KASH5 and dynein is disrupted by mutation of the KASH5 EF-hand domain, although dynein recruitment to KASH5 in cells is calcium-independent. Since defective synapsis can lead to genetic abnormalities and infertility, the characterization of the KASH5–dynein interaction is an important step forward in understanding the complex mechanism of chromosome movement during mammalian meiotic prophase I.

## Results

### KASH5 forms a complex with dynein, dynactin, and LIS1
As KASH5 is expressed only in the meiotic germ line, HeLa cells provide a convenient "KASH5-null" background for studying the interaction between dynein and KASH5 (Horn et al., 2013; Morimoto et al., 2012). We generated a stable HeLa cell line in which expression of GFP-KASH5 was induced by doxycycline so that the recruitment of endogenous dynein and dynactin to the nuclei of KASH5-expressing cells could be visualized by immunofluorescence. LIC1 and dynactin p50 were both recruited to KASH5-expressing nuclei (Fig. 1 A), in addition to dynactin p150 and IC (Horn et al., 2013). In contrast, neither dynein nor dynactin was recruited to the nuclei of cells transiently expressing a different KASH protein, nesprin-2α2 (GFP-N2α2). Recruitment of dynein to the nuclear envelope in untransfected cells was rare since it only occurs in late G2-prophase (Baffet

et al., 2015; Hu et al., 2013; Salina et al., 2002; Splinter et al., 2012; Splinter et al., 2010), and examples where LIC1 is not seen at the NE can be seen in Fig. 6 B, Fig. 8 C, and Fig. S5 A.

We confirmed the recruitment of dynein and dynactin to KASH5 biochemically. HeLaM cells were transiently transfected with GFP-KASH5ΔK (Fig. 1 B) or GFP-N2α2ΔK, which lack their KASH and transmembrane domains, and so are cytosolic. LIC1, LIC2, IC, and p150 are all associated with GFP-KASH5ΔK and not with GFP-N2α2ΔK (Fig. 1 C). The KASH5 N-terminal 166 amino acids (GFP-KASH5ND), containing the EF-hands, was able to pull down dynein IC and dynactin from cell lysates (Horn et al., 2013), and we found that it was somewhat more efficient than GFP-KASH5ΔK for LIC1 (Fig. 1 D), as seen for IC (Horn et al., 2013).

LIS1 plays a key role in the assembly of DDA and D₂DA complexes (reviewed in Canty and Yildiz, 2020; Markus et al., 2020; Olenick and Holzbaur, 2019) and coprecipitates with BICD2N, dynein, and dynactin (Splinter et al., 2012). As reported previously (Horn et al., 2013), GFP-KASH5ΔK pulled down LIS1 as well as dynein and dynactin, but this complex excluded the dynein adaptor BICD2 (Fig. 1 C). Immunofluorescence analysis revealed that endogenous LIS1 was detected at the NE in 99.8% of KASH5-expressing cells (Fig. 1 A and Fig. S1 C), whereas the LIS1 and dynein interactor Nde1 was not, even though it could be detected at the NE in late G2 cells (Fig. 1 A).

### Hierarchy of dynein, dynactin, and LIS1 recruitment to KASH5
We next sought to identify which dynein subunits mediate the KASH5 interaction and the roles played by dynactin and LIS1. We used RNA interference to deplete individual dynein subunits in the GFP-KASH5 cell line (Fig. S2 A) and assessed the effect on recruitment to KASH5. Depletion of dynein IC2 (the only form expressed in HeLa cells [Palmer et al., 2009]) did not prevent dynein recruitment (Fig. 2 A), with 100% of KASH5-positive nuclei being labeled with anti-LIC1 (Fig. S1 C). Cytosolic levels of LIC1 were reduced following IC2 depletion, making the NE pool particularly distinct. This is likely due to a modest reduction in total dynein levels when IC2 is depleted, as seen by immunoblotting with anti-LIC1 and 2 (Fig. S2 A). LIS1 recruitment to KASH5 at the NE was also unaffected by the depletion of IC2 (Fig. 2 A and Fig. S1 C). Strikingly, dynactin was only rarely detected at the NE in IC2-depleted cells using antibodies to p150 or p50 (Fig. 2 A and Fig. S1 C), even though p150 was still readily observed at microtubule plus ends. These data suggest that while the interaction between IC and p150 (Karki and Holzbaur, 1995; King et al., 2003; Vaughan and Vallee, 1995) is not needed for dynein and LIS1 recruitment to KASH5, it is essential for the association of dynactin with KASH5.

Since IC2 depletion reduced total cellular dynactin p150 levels by ~25% (Fig. S2 A), we used a dominant negative approach as another way of testing the effect of disrupting IC2–p150 interactions on recruitment to KASH5 using overexpression of the coiled coil 1 region of p150 (CC1; King et al., 2003; Quintyne et al., 1999). Unfortunately, transiently transfected constructs would not coexpress with GFP-KASH5 in the inducible GFP-KASH5 cell line. We therefore transiently cotransfected HeLa cells with GFP-KASH5 and myc-SUN2 (to

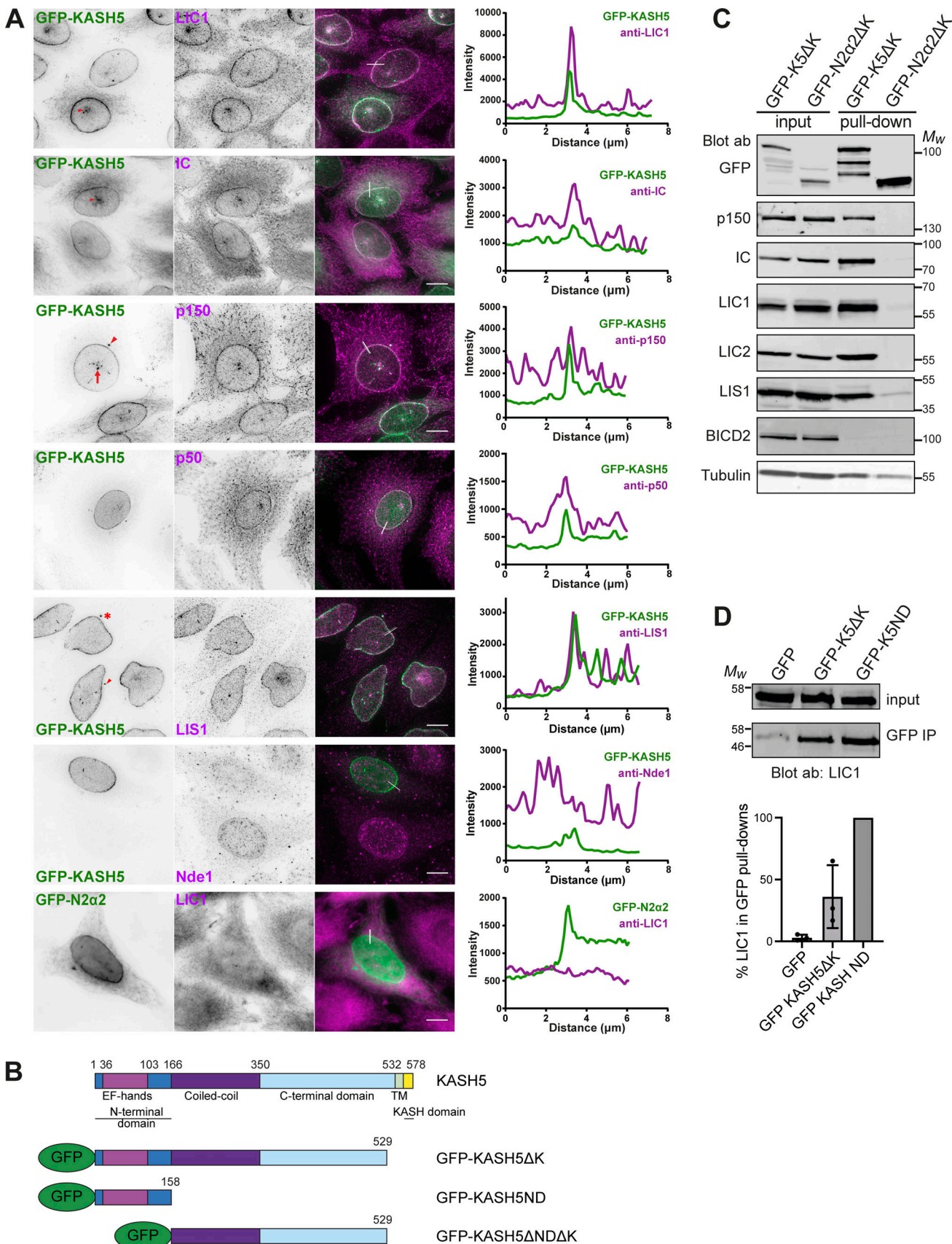

**Figure 1. Recruitment of dynein, dynactin, and LIS1 to KASH5 in HeLa cells. (A)** A stable HeLa cell line inducibly expressing GFP-KASH5 (green) was labeled with antibodies against dynein IC and LIC1, dynactin p150 and p50, LIS1, and Nde1 (magenta) and imaged on a DeltaVision microscope. Images are

z-projections of deconvolved image stacks. The bottom panel shows the transient expression of GFP-N2α2 in green and labeling with anti-LIC1 in magenta, with undeconvolved wide-field images. Arrowheads point out the location of centrosomes, full arrows show creases in the NE, and asterisks mark cytoplasmic accumulations of GFP-KASH5. Thin white lines on color merge images show where a line scan plot was performed, shown on the right. Scale bars = 10 µm. **(B)** Schematic showing KASH5 and the constructs used. For some experiments, the GFP was replaced with an HA tag. **(C)** Dynein, dynactin, and LIS1 are recruited to KASH5 as shown by GFP-Trap pull-downs. HeLaM cells were transiently transfected with GFP-KASH5ΔK or GFP-N2α2ΔK. The pull-downs and inputs (1.5% of total lysate) were probed with antibodies against GFP, p150, IC, LIC1, LIC2, LIS1, BICD2, and α-tubulin. Molecular weight markers are shown. Quantitation of the blot is given in Table S1. **(D)** The KASH5 N-terminal EF-hand domain is sufficient to recruit dynein. Lysates of HeLaM cells expressing GFP, GFP-KASH5ΔK (GFP-K5ΔK), or GFP-KASH5 N terminus (GFP-K5ND) were isolated by GFP-trap and probed for LIC1 by immunoblotting. The input is 15% of the GFP-trap sample. The graph shows the quantitation of LIC1 levels in pull-downs (n = 3 independent experiments), normalized to GFP-K5ND. Error bars denote SD. Source data are available for this figure: SourceData F1.

enhance localization of KASH5 to the NE) along with RFP-tagged CC1. Notably, while CC1 expression had no effect on dynein or LIS1 recruitment to KASH5, it prevented dynactin accumulation at the NE (Fig. 2 B), confirming that dynein and LIS1 can associate with KASH5 independently of dynactin. As SUN1 and SUN2 have been shown to play a role in Golgi apparatus position and organization (Hieda et al., 2021), the expression of additional SUN2 could potentially alter Golgi apparatus morphology. However, we did not use coexpression of SUN2 and KASH5 in any assays of organelle positioning.

We used RNAi to investigate the involvement of LICs in the dynein–KASH5 interaction. There was no reduction in the proportion of cells with detectable dynein, dynactin, or LIS1 recruited to KASH5 after depleting LIC2 alone (Fig. 3, B and D). LIC1 depletion also had very little effect on dynactin or LIS1 recruitment in a binary scoring assay but had a variable effect on the dynein intermediate chain, with 80.8 ± 24.3% of cells (n = 3 experiments, ±SD) showing IC signal at the NE (Fig. 3, A and D). However, when both LIC1 and LIC2 were depleted simultaneously, the proportion of cells with detectable dynein and dynactin at the NE was reduced by ~75% (Fig. 3, C and D). LIS1 recruitment was also reduced, with ~40% of KASH5 expressing cells showing no LIS1 signal at the NE. We were not able to deplete endogenous LIC1 completely using RNAi (Fig. S2 B), which may explain why in some cells a residual level of dynein, dynactin, and LIS1 remained with KASH5. In addition, GFP–KASH5ΔK pull-downs from LIC1 and 2 depleted HeLa cells contained very little dynein and dynactin (Fig. 3 E). Thus, LIC1 and 2 act redundantly to recruit dynein and LIS1 to KASH5 at the NE, with dynactin being recruited downstream, by a mechanism requiring the interaction between IC and p150.

To test if LIS1 is required for KASH5 to associate with dynein and dynactin, we depleted LIS1 using RNAi. Interestingly, the proportion of cells with LIC1 detectable at the NE was virtually unaffected by LIS1 knockdown (Fig. S1, D and E) whereas p150 recruitment was seen in only 20% of cells (Fig. S1, D and G). Dynein IC detection at KASH5-positive nuclei varied between experiments (Fig. S1, D and F), with a mean of 57.1% of cells positive for IC74 antibody labeling. The discrepancy between the anti-LIC1 and IC scoring most likely reflects the weaker NE labeling seen with the IC74 antibody compared with anti-LIC1 antibodies in control cells. Our interpretation of these data is that LIS1 depletion reduces dynein levels at the NE somewhat to levels that are still detectable by anti-LIC1 but sometimes not by IC74. Altogether, these data reveal that LIS1 is vital for the dynactin complex to be recruited to KASH5 downstream of dynein

and suggest that LIS1 may also promote the formation of or stabilize the KASH5–dynein complex.

**Dynein LIC1 helix 1 is essential for KASH5 binding**

We next wanted to ascertain the LIC1 region responsible for the dynein–KASH5 interaction. Assays with purified protein have shown that helix 1 and some preceding amino acids (433–458) formed the minimal LIC1 region needed for interaction with Hooks, BICD2, and Spindly (Lee et al., 2018), with additional interactions between LIC1 and Spindly, BICD2, and Hook3 being seen in helix 2 and the linker region upstream of helix 1 (Celestino et al., 2019). In contrast, RILP required only helix 1 (LIC1 440–455; Celestino et al., 2019). We made a series of hLIC1 constructs (Fig. 4 A): full-length LIC1, a truncation containing helix 1 but not helix 2 (LIC1-CT2: amino acids 1–456) and a construct that terminates shortly after the Ras-like domain and lacks both helices and the linker sequences (LIC1-CT3: amino acids 1–388). Cells depleted of LIC1 and 2 by RNAi were co-transfected with myc-tagged LIC1 constructs and the soluble GFP-KASHND. GFP trap IP revealed that while dynein containing LIC1-FL and LIC1-CT2 was recruited to the KASH5 N-terminal domain, dynein with LIC1-CT3 was not (Fig. 4 B). Previous work has shown that deletion of just helix 1 (N440-T456) or helix 1 point mutants (F447A/F448A and L451A/L452A) prevented binding of the LIC1 C-terminal domain to several dynein adaptors using purified proteins (Celestino et al., 2019). We generated the helix 1 deletion and mutations in GFP-LIC1 (Fig. 4 A) and also a deletion lacking helix 1 plus some linker sequence (S433-S458). These were coexpressed with HA-tagged KASH5ΔK in HeLaM cells depleted of both LICs. Full-length GFP-LIC1 pulled down KASH5ΔK effectively, whereas the LIC1 helix 1 deletions and point mutants did not (Fig. 4 C). We then examined the recruitment of LIC1 to KASH5 at the NE in HeLaM cells coexpressing HA-KASH5, myc-SUN2, and GFP-LIC1 constructs. Both wild-type GFP-LIC1 and the mid-length LIC1-CT2 were recruited to KASH5, although CT2 detection at the NE was variable (Fig. 4, D and E). In contrast, LIC1-CT3, the helix 1 deletions, and helix 1 point mutants were poorly recruited. Altogether, these results demonstrate LIC1 helix 1 is essential for interaction with KASH5.

The importance of LIC helix 1 in cargo binding prompted us to assess the role of the LICs on endogenous membrane cargos whose adaptors have not been fully defined. First, we tested if LIC1 and 2 functioned redundantly in Golgi apparatus morphology and positioning, for which there is conflicting published evidence (Kumari et al., 2021a; Palmer et al., 2009; Tan et al.,

## A  IC2 siRNA

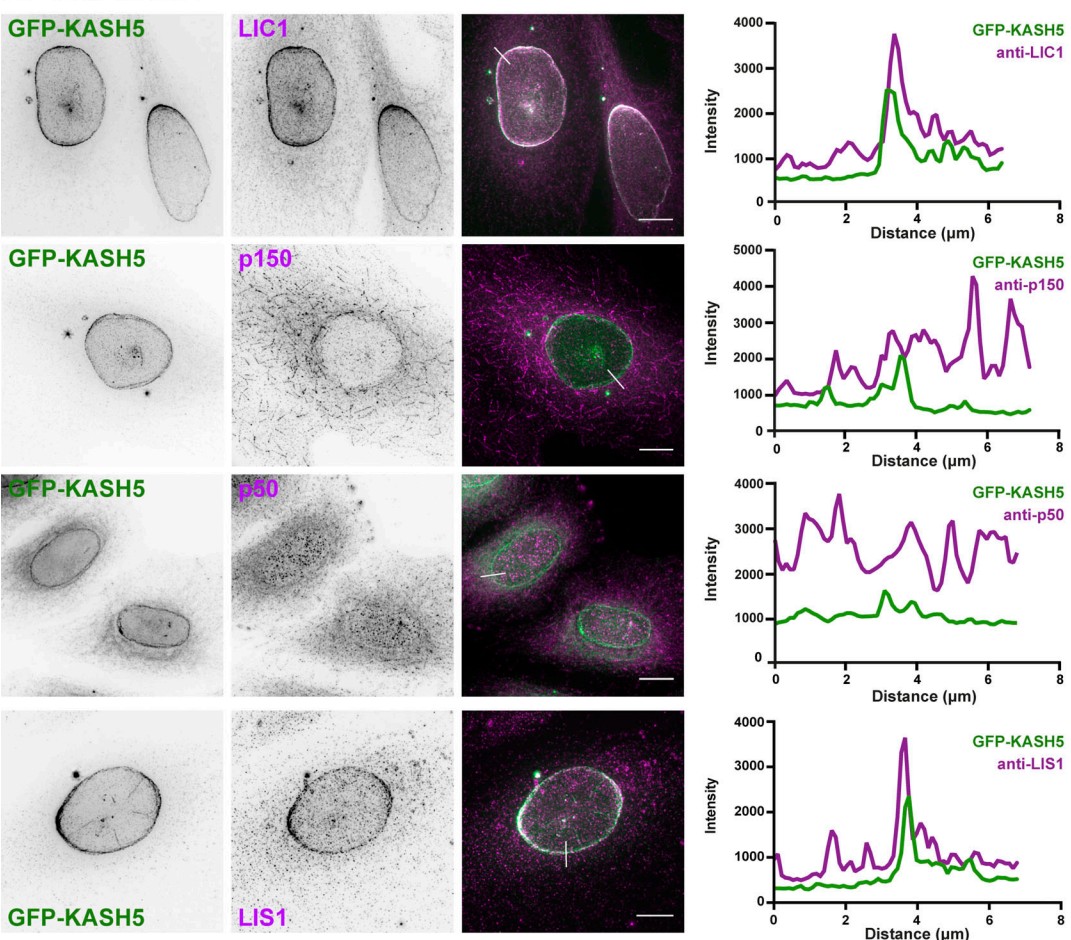

## B  RFP-p150-CC1 expression

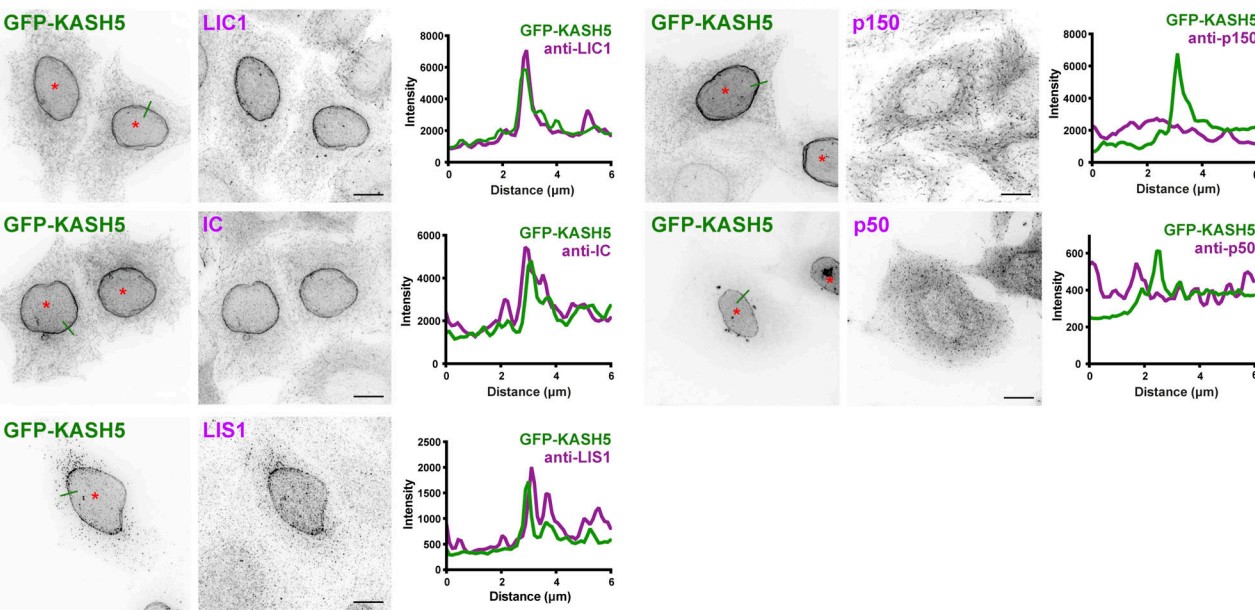

**Figure 2. Dynein and LIS1 recruitment to KASH5 does not require interaction between dynein IC and dynactin p150. (A)** HeLa cells stably expressing GFP-KASH5 (green) were depleted of IC2 using 20 nM siRNA for 72 h and then processed for immunofluorescence with antibodies against LIC1, dynactin p150, and LIS1 (magenta). White lines on color merge images show where a line scan plot was performed, shown on the right. **(B)** HeLa cells were transiently transfected with GFP-KASH5, RFP-CC1, and myc-SUN2 (CC1 and SUN2: not shown. Cells expressing CC1 are marked with asterisks). Cells were fixed and

labeled with antibodies against dynein IC and LIC1, dynactin p50 and p150, and LIS1. Green lines on the GFP-KASH5 images indicate line scan locations, shown on the right. Images were taken on a DeltaVision microscope and z-stack projections of deconvolved images are shown. Scale bars = 10 µm.

2011). RNAi depletion of LIC1 or 2 individually led to the break-up of the Golgi ribbon in ~60% of HeLaM cells, with the Golgi fragments remaining centrally located (Fig. 5, A–C). In contrast, the depletion of both LICs simultaneously led to complete fragmentation and scattering of the Golgi apparatus (Fig. 5, B and C), demonstrating that the LICs act redundantly in Golgi positioning. This was confirmed by expressing RNAi-resistant LIC1-mKate or LIC2-mKate in cells depleted of both LICs (Fig. 5 D), as either LIC was able to fully restore Golgi apparatus clustering. The endocytic pathway also relies heavily on dynein activity (Flores-Rodriguez et al., 2011; Granger et al., 2014; Reck-Peterson et al., 2018; Wang et al., 2019), which contributes to the sorting of endocytic cargo in the early endosome (Driskell et al., 2007). Depletion of both LICs profoundly altered the distribution of early endosomes, recycling endosomes, and lysosomes (Fig. S3 A), while only minimal effects were seen with single depletions (not shown). Moreover, the expression of either LIC1-mKate or LIC2-GFP restored the position of endocytic organelles (Fig. S3 B), confirming that LICs act redundantly in this context. We investigated if the same effects were observed with dynein recruitment to RILP. Recruitment of dynein and dynactin to HA-RILP-positive late endosomes was significantly reduced when both LICs were depleted together, but not individually (Fig. S4, A–D).

To examine if the LIC1 region 388–456 that contains helix 1 is needed for Golgi apparatus and endocytic organelle positioning, we used our knock-down and rescue approach (Fig. 5 E, Fig. S3 C and D, and Fig. S4 E). Both full-length GFP-LIC1 and GFP-LIC1-CT2 rescued the Golgi apparatus, early endosome, and lysosome clustering. In contrast, GFP-LIC1-CT3 did not, with EEA1-positive early endosomes (Fig. S3 C) and LAMP1-positive late endosomes/lysosomes (Fig. S3 D) remaining localized in the cell periphery, and the Golgi apparatus was fragmented and scattered (Fig. 5 E). Moreover, GFP-LIC1-CT3 did not interact with RILP, whereas GFP-LIC1-CT2 or full-length GFP-LIC1 were robustly recruited to RILP-positive organelles (Fig. S4 E).

Altogether, these data provide clear in cellulo evidence that the LIC adaptor binding helix 1 is needed for dynein's interaction with KASH5 and RILP and is also crucial for dynein's function on the Golgi apparatus, early endosomes, and lysosomes. Furthermore, we demonstrate that LIC1 and LIC2 act redundantly in these situations.

### KASH5 is a novel activating dynein adaptor
KASH5 shares key properties with other activating dynein adaptors: interaction with LIC helix 1; ability to recruit dynactin; and presence of an N-terminal dynein binding domain (containing EF-hands) followed by an extended coiled coil (Horn et al., 2013). In addition, KASH5's biological function in telomere clustering in the prophase of meiosis I (Horn et al., 2013; Lee et al., 2015) strongly suggests that it recruits active dynein. When expressed out of its meiotic context, in HeLa cells, we quite often observed clusters of KASH5 and dynein around

discrete points close to or on top of the nucleus (red arrowheads and arrows in Fig. 1 A), suggestive of clustering around the centrosome via active dynein. In addition, the asymmetric distribution of KASH5 in the NE with enrichment toward the MTOC has been noted (Horn et al., 2013).

Overexpression of activating dynein adaptor proteins lacking their cargo binding domain disrupts dynein functions by sequestering dynein and preventing its binding to endogenous adaptors (e.g., Hoogenraad et al., 2001; Hoogenraad et al., 2003; Horgan et al., 2010a; Horgan et al., 2010b; Splinter et al., 2012). As KASH5 interacts with LIC helix 1, we hypothesized that it would compete with other adaptors for binding to dynein. Indeed, overexpression of cytosolic GFP-KASH5ΔK in HeLaM cells resulted in complete fragmentation of the Golgi apparatus and redistribution of lysosomes to the cell periphery; phenotypes indicative of a loss of dynein function (Fig. 6 A). In contrast, the expression of a cytoplasmic KASH5 construct lacking its N terminal dynein binding domain (Fig. 1 B, GFP-KASH5ΔNDΔK [Horn et al., 2013]) had no effect on Golgi apparatus or lysosome distribution and morphology (Fig. 6 A). The effects of GFP-KASH5ΔK are therefore due to KASH5 binding and sequestering dynein, preventing its recruitment to other membrane cargoes.

We next determined if KASH5 competes with established dynein adaptors for dynein binding. To test this possibility, we cotransfected HeLaM cells with HA-tagged full-length KASH5, myc-tagged SUN2, and dominant negative versions of the activating adaptors BICD2 (GFP-BICD2N [Hoogenraad et al., 2001; Splinter et al., 2012]) and Rab11-FIP3 (GFP-Rab11-FIP3 I737E, which retains its LIC binding domain but is unable to interact with Rab11 [Wilson et al., 2005]). Endogenous dynein was recruited to KASH5 at the NE in control cells, but this was prevented when dominant-negative dynein adaptors BICD2N and Rab11-FIP3 I737E were expressed (Fig. 6 B). We investigated if the same was true for dynein recruitment to RILP, even though RILP is not thought to be able to activate dynein/dynactin motility (Lee et al., 2020; Reck-Peterson et al., 2018). Dynein was recruited to RILP-positive organelles following overexpression of HA-tagged RILP (Fig. 6 C). However, when GFP-BICD2N or GFP-Rab11-FIP3 I737E were coexpressed, recruitment of dynein LIC1 to RILP was abrogated (Fig. 6 C). Taken together, these findings show that established dynein adaptor proteins compete with KASH5 and RILP for dynein binding.

To test directly if KASH5 could act as an activating adaptor and form motile complexes with dynein and dynactin, we used single molecule in vitro motility assays consisting of purified bacterially expressed KASH5 truncations (Fig. 7 A) mixed with purified fluorescently labeled dynein, dynactin, and LIS1. We generated three KASH5 constructs encoding amino acids 1–407, 1–460, and 1–507, all of which contain the N-terminal EF-hand dynein binding domain plus the predicted coiled-coil region (amino acids 166–350), plus a variable amount of the C-terminal region. SEC-MALS analysis showed that all three proteins formed dimers (Fig. 7 B). Of these, KASH5$_{1-460}$ was the most

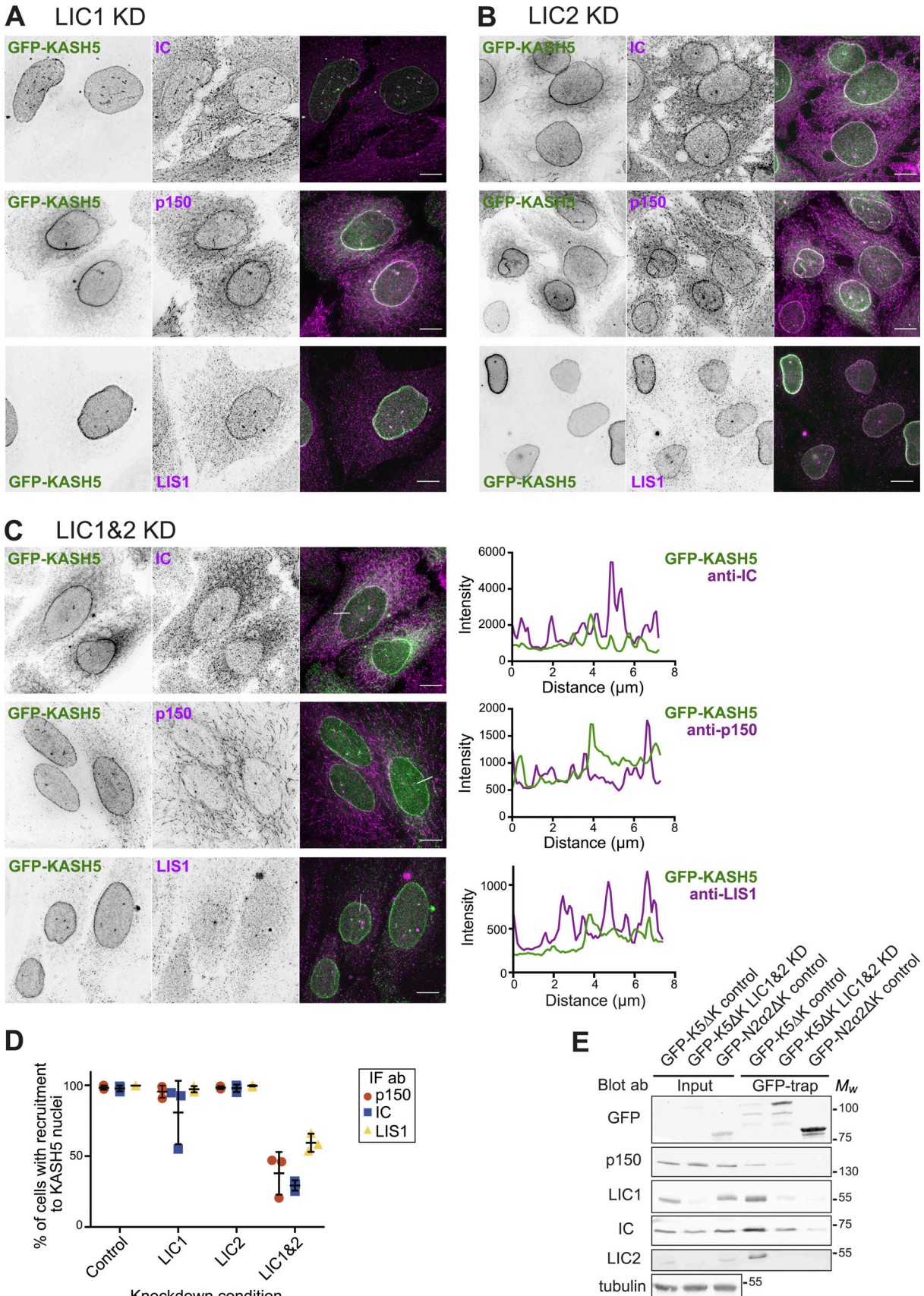

Figure 3. **Dynein is recruited to KASH5 via either LIC1 or 2. (A–D)** HeLa cells stably expressing GFP-KASH5 (green) were depleted of LIC1 (A), LIC2 (B), or both LIC1 and 2 (C) using 10 nM of each siRNA for 72 h, then fixed, and labeled with antibodies against IC, dynactin p150, or LIS1 (magenta). Thin white lines

show where a line scan plot was performed on LIC1 and 2 depleted cells, shown on the right. Images were taken on a DeltaVision microscope and z-stack projections of deconvolved images are shown. Scale bars = 10 μm. **(D)** LIC-depleted cells were scored in a binary fashion for recruitment of IC, p150, or LIS1 to KASH5. The mean and standard deviation of each condition are shown. The experiment was repeated three times, with 300 cells scored for each condition in each experiment. **(E)** HeLaM cells were depleted with siRNA against both LICs together (10 nM each) or with control siRNAs. 48 h into the knockdown, cells were transfected with either GFP-KASH5ΔK or GFP-N2α2ΔK. The following day cells were lysed and a GFP trap was performed. Input and pull-downs were immunoblotted with antibodies against GFP, p150, LIC1, LIC2, IC, and α-tubulin. The input was 1.5% of the total cell lysate. Molecular weight markers are shown on the right. Quantitation of the blot is given in Table S1. Source data are available for this figure: SourceData F3.

active in motility assays using dynein containing a heavy chain mutant that cannot form the inhibited Phi conformation (Zhang et al., 2017; Fig. 7 C). We next used assays with wild-type purified dynein, which when combined with dynactin and LIS1 alone displayed very little motility (Fig. 7 D). The inclusion of KASH5$_{1-460}$ promoted processive dynein movements (Fig. 7, D and E), although a lesser number than seen with purified Hook3$_{1-522}$ (Urnavicius et al., 2018). KASH5 EF-hands (amino acids 1–115), the coiled coils (amino acids 155–349), or a combination of EF-hands and coiled coils alone (amino acids 1–349) were not enough to activate dynein/dynactin (Fig. 7 F). Importantly, the velocity of processive dynein movements was the same for KASH5$_{1-460}$ and Hook3$_{1-522}$ (Fig. 7 E). KASH5, a transmembrane protein, is therefore an activating dynein adaptor.

### KASH5's EF hand is critical for dynein and dynactin complex assembly

A common feature of several dynein adaptor proteins is the presence of an N terminal pair of EF-hands in the dynein-binding domain, as seen in Rab11-FIP3, CRACR2a, Rab45/RASEF, and ninein (Celestino et al., 2019; Lee et al., 2020; Reck-Peterson et al., 2018; Wang et al., 2019). KASH5 contains two putative EF hands, extending from amino acids 36–103, which form the bulk of the dynein binding domain (Fig. 1, B and D; and Fig. 4 C). Sequence alignment of the KASH5 EF hands with CRACR2a, Rab45, FIP3, and ninein revealed that KASH5, like FIP3 and ninein, lacks some key consensus calcium-binding amino acids (at positions X,Y,Z,-X,-Y,-Z), with these changes being consistent across species (Fig. 8 A). EF-hand 1 is particularly divergent, with only the residue in position X (Grabarek, 2006) matching the consensus, and with the key position -Z being a glutamine or histidine instead of a glutamate residue. While EF-hand 2 has consensus amino acids in position X, other residues either do not conform or vary between species. For example, in non-rodent KASH5 EF-hand 2, there is aspartate in place of glutamate at -Z, which can result in magnesium binding (Grabarek, 2006), as suggested for FIP3 (Lee et al., 2020).

This analysis suggested that KASH5 might not be calcium regulated. To determine if the KASH5 and Rab11–FIP3 interaction with dynein in cells required calcium or not, we expressed GFP-tagged KASH5 with myc-SUN2 in HeLaM cells, or GFP-Rab11-FIP3 in Vero cells, and labeled for endogenous LIC1. In control DMSO-treated cells, there was robust recruitment of dynein to KASH5 at the NE and to Rab11-FIP3-positive recycling endosomes (Fig. S5, A and B). This recruitment was not affected by treating cells with the cell-permeable calcium chelator, BAPTA-AM, for 2 h to deplete intracellular calcium. Furthermore, endogenous LIC1 was present at equal levels when GFP-KASH5-ND was pulled down from cell lysates in the presence or

absence of EGTA (Fig. S5 C). This demonstrates that the KASH5–dynein interaction does not require calcium and confirms that the Rab11–FIP3–dynein interaction in cells is calcium-independent, as reported for in vitro assays (Lee et al., 2020).

We wanted to test the importance of KASH5's EF-hands in the dynein interaction, even if calcium was not required. The *fue* mutation in the zebrafish KASH5 homolog, futile cycle (fue; Lindeman and Pelegri, 2012), gives zygotes that are defective in pronuclear migration and mitotic spindle assembly (Dekens et al., 2003; Lindeman and Pelegri, 2012). The corresponding mutation in human KASH5 changes a valine to glutamic acid in EF-hand 1 (V54E: Fig. 8 A). We generated KASH5-EF-*fue* constructs to establish how this mutation affected KASH5–dynein interactions. We also mutated the amino acids in positions X and Y of both EF-hands to alanine (KASH5-EF-AA: D44A; Q46A; D81A; N83A: Fig. 8 A), as these mutations in CRACR2a ablate its function (Srikanth et al., 2016; Wang et al., 2019). Lastly, we replaced some KASH5 residues with those found in CRACR2a (Fig. 8 A). KASH5-EF-mod1 has four substitutions: Q46E, Q55D, P87Y, and K88L. In KASH5-EF-mod2, nine amino acids in EF-hands 1 and 2 and part of the exiting helix were changed to the CRACR2a sequences.

To test which KASH5 mutants could form a stable complex with dynein and dynactin, GFP-KASH5ΔK-WT or EF-hand mutants were isolated from HeLaM cells by GFP-trap and probed for endogenous IC and dynactin p150. Unmodified GFP-KASH5ΔK coprecipitated with dynein and dynactin, as did GFP-KASH5ΔK-EF-mod1 (Fig. 8 B). Interestingly, dynein recruitment to EF-mod1 was greater than to KASH5ΔK-WT in 2/3 experiments. In contrast, much less dynein and dynactin bound to the GFP-KASH5ΔK-EF-*fue*, EF-AA, or EF-mod2 mutants. In agreement, we found that endogenous dynein was recruited to full-length GFP-KASH5 and GFP-KASH5-EF-mod-1 at the NE, whereas GFP-KASH5-EF-*fue*, EF-AA, or EF-mod2 mutants did not accumulate dynein (Fig. 8, C and D).

As another means of assessing the effects of these mutations on KASH5–dynein binding, we harnessed the dominant negative effect of expressing cytosolic KASH5, which causes Golgi fragmentation by sequestering dynein (Fig. 6 A). In this assay, any mutant that prevents KASH5 from binding dynein would have no effect on Golgi morphology when expressed, as seen with GFP alone (Fig. 8 E). Overexpression of GFP-KASH5ΔK or GFP-KASH5ΔK-mod1 in HeLaM cells resulted in strong fragmentation of the Golgi apparatus. In contrast, GFP-KASHΔK-EF-*fue*, GFP-KASHΔK-EF-AA, or GFP-KASHΔK-EF-mod2 (Fig. 8 E) had a much weaker effect on Golgi positioning, implying that these EF-hand mutants were much less able to sequester dynein. Altogether, these findings show that the KASH5 EF-hand is critical for its function with dynein and dynactin, although the interaction is not calcium-dependent.

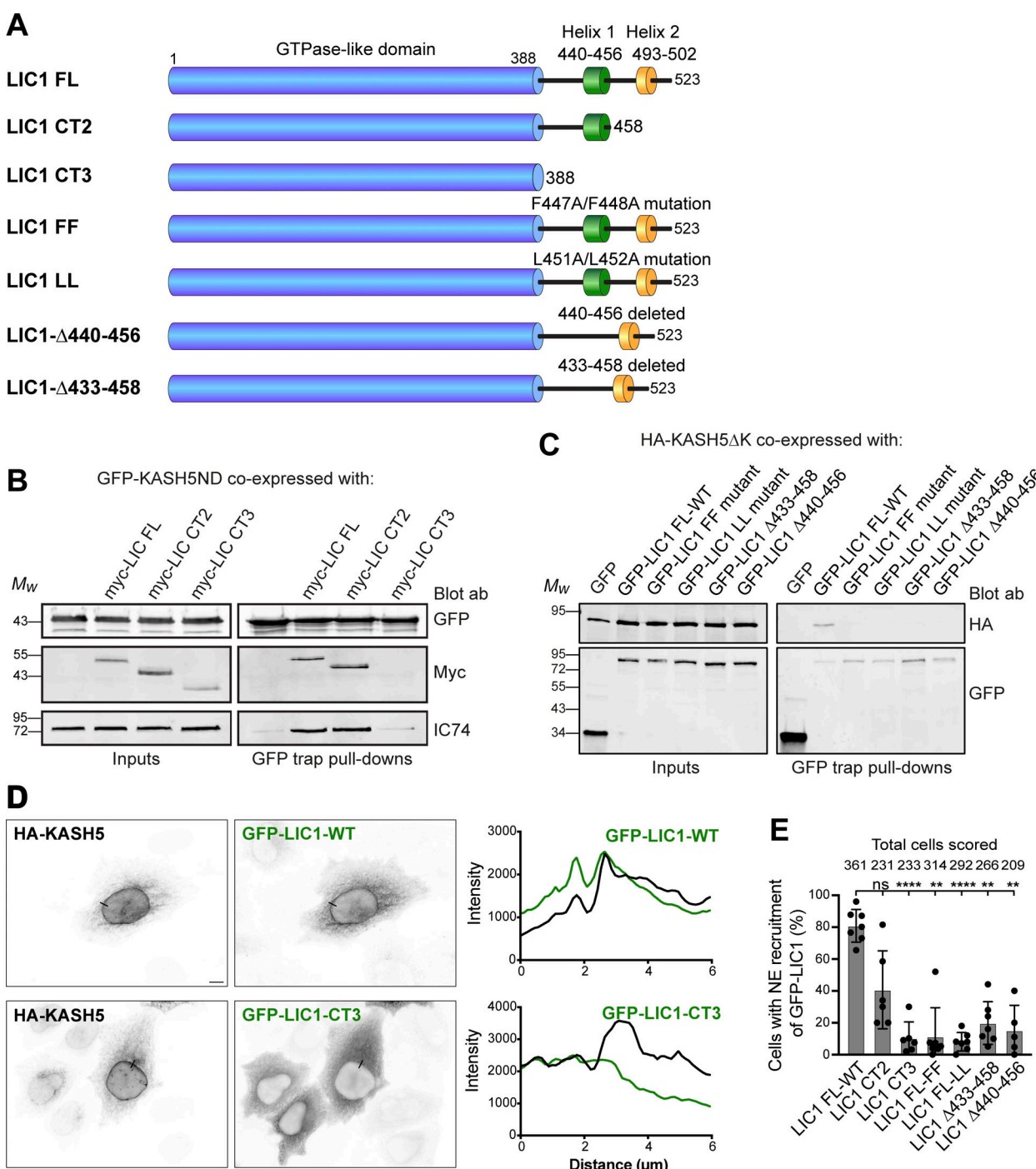

**Figure 4. KASH5-LIC1 interactions require LIC1 helix 1. (A)** Schematic of LIC1 showing the GTPase-like domain that is highly conserved between LICs, and a less well conserved C terminal domain containing helix 1 (440–456) and helix 2 (493–502). The constructs used are: truncations lacking helix 2 (LIC1 CT2) or both helix 1 and helix 2 (LIC1 CT3); point mutations within Helix 1 (FF: F447A/F448A and LL: L451A/L452A); helix 1 deletions (Δ440–456 and Δ433–458). **(B and C)** HeLaM cells were depleted of both LICs by siRNAs for 48 h, then cotransfected, and incubated for a further 24 h with Myc-tagged LIC1 constructs and GFP-KASH5ND or GFP as a control (B) or GFP-LIC1 constructs and HA-KASH5ΔK (C). Cell lysates were incubated with GFP-trap beads and then analyzed by SDS-PAGE and immunoblotting with antibodies to GFP, myc, and IC (IC74) to detect native dynein, as indicated. The input is 15% of the GFP-trap sample. Quantitation of myc-LIC1 pull-down efficiency (B) with GFP-KASH5ND vs. myc-LIC1-FL was 177% for LIC1-CT2 and 8.0% for LIC1-CT3 (n = 2). Quantitation of HA-KASH5DK pull-down efficiency (C) vs. GFP-LIC1-WT was 3.9 ± 3% for LIC1-FF, 3.3 ± 4.9% for LIC1Δ433-458 and 1.9 ± 2.1% for LIC1Δ440-456 (± SD, n = 3) and 3.1% for LIC1-LL (n = 2). **(D and E)** HeLaM cells were co-transfected with GFP-tagged LIC1 constructs, HA-KASH5, and myc-SUN2 for ~18 h, then fixed, and labeled with antibodies to HA, GFP, and myc (not shown) and imaged by wide-field microscopy. Cells with strong enrichment of KASH5 and SUN2 at the NE were scored for the presence or absence of GFP at the NE without knowing the identity of the LIC1 construct expressed (see Materials and methods). **(D)** Example wide-field images with line scans (black lines on the images): GFP-LIC1 constructs shown in green, HA-KASH5 in black. GFP-LIC1-WT is recruited to HA-KASH5 (top panels) whereas GFP-LIC1-CT3 is not (bottom panels). Scale bar = 10 µm. **(E)** Recruitment of GFP-LIC1 proteins to the NE was scored in a binary fashion from five to seven experiments with 25–82 cells scored per experiment per construct. The mean ± SD is plotted, and the total

number of cells scored for each condition is shown above the line. A mixed effects analysis was performed (see Materials and methods) with Tukey's multiple comparison test to compare GFP-LIC1 FL recruitment vs. all others (P ≤ 0.0001 = ****, ≤0.001 = ***, ≤0.01 = **). No other comparisons gave significant P values. Source data are available for this figure: SourceData F4.

## Discussion

Infertility will affect ∼1 in 7 couples trying to conceive and has extraordinarily detrimental effects on those affected. Despite this, surprisingly little is known about the multitude of molecular and genetic causes of infertility. This is largely due to the complexity of meiosis and pronuclear migration and a lack of samples to study from sterile populations. Dynein generates the mechanical force required for the dynamic chromosome movements in meiotic prophase I that are essential for meiotic progression and maintaining genetic integrity. KASH5 is part of the mammalian LINC complex component that spans the NE to link dynein to telomeres and is essential for synapsis and meiotic progression (Horn et al., 2013; Morimoto et al., 2012). Here, we show that KASH5 is a transmembrane activating adaptor for dynein, in agreement with a recent report (Agrawal et al., 2022). We reveal that dynein–KASH5 interactions require LIC helix 1 (LIC1 residues 440–456), which mediates dynein's interaction with many other cargo adaptors (Celestino et al., 2022; Celestino et al., 2019; Lee et al., 2020; Lee et al., 2018). Accordingly, KASH5 competes with established dynein adaptors for dynein binding, and expression of a cytosolic KASH5 truncation inhibits dynein interphase function (Fig. 6), as seen for other dynein adaptor constructs that cannot bind cargo (Hoogenraad et al., 2001; Hoogenraad et al., 2003; Horgan et al., 2010a; Horgan et al., 2010b; Splinter et al., 2012). Our in vitro assays (Fig. 7) confirm that KASH5 is an activating dynein adaptor (McKenney et al., 2014; Schlager et al., 2014a; Schlager et al., 2014b) as KASH5, in the presence of LIS1 and dynactin, promotes motility of purified dynein molecules (Fig. 7), as recently reported (Agrawal et al., 2022).

Considerable force must be exerted to move chromosomes within the nucleoplasm to promote synapsis. There may be ∼80 LINC complexes per telomere (Spindler et al., 2019), each providing a point where dynein motors can engage with the cytoplasmic face of the NE. A key question is how many dyneins bind to each LINC complex, which likely contains three KASH5 dimers bound to two SUN trimers (Gurusaran and Davies, 2021). Solution studies suggest that a single LIC protein binds per KASH5 dimer (Agrawal et al., 2022), in keeping with the report that two BICD2 proteins can assemble with one dynactin and two dyneins (Chaaban and Carter, 2022), with the LICs from one of the dyneins interacting with both of the BICD2 adaptors. This suggests that there may be one or two dyneins per LINC complex and an ensemble of 80–160 dyneins per telomere.

Importantly, we find that KASH5 recruits LIS1 as well as dynein and dynactin. LIS1 is crucial for dynein function where high force is needed (Chapman et al., 2019; Markus et al., 2020; Pandey and Smith, 2011; Reddy et al., 2016; Yi et al., 2011). It increases dynein force generation in vitro by promoting the recruitment of two dynein motors per dynactin (Elshenawy et al., 2020; Htet et al., 2020; Markus et al., 2020) and opens the dynein phi complex to allow easier assembly of the dynein/ dynactin/adaptor complex (Elshenawy et al., 2020; Gillies et al., 2022; Htet et al., 2020; Marzo et al., 2020; Qiu et al., 2019). Indeed, in cells, LIS1 enhances dynein and dynactin recruitment to a wide range of cellular cargoes, including the NE in late G2 (Cockell et al., 2004; Dix et al., 2013; Dzhindzhev et al., 2005; Lam et al., 2010; Siller et al., 2005; Sitaram et al., 2012; Splinter et al., 2012; Wang et al., 2013). Our data suggest that LIS1 is essential for recruiting dynactin to KASH5 (Fig. S1). Surprisingly, LIS1 depletion had much less effect on dynein recruitment. We also saw that interfering with dynein IC–p150 interactions by IC2 depletion or over-expression of p150 CC1 did not prevent dynein recruitment to KASH5. Based on these data, we propose that the first step in KASH5 adaptor complex assembly is an interaction between LIC helix 1 and the KASH5 EF-hands. Next, the binding of LIS1 opens the dynein phi complex. The third step involves the recruitment of the dynactin complex initiated by the IC-p150 interaction, followed by the formation of extensive contacts between the dynactin complex and the adaptor (Chowdhury et al., 2015; Grotjahn et al., 2018; Lau et al., 2021; Lee et al., 2020; Urnavicius et al., 2018; Urnavicius et al., 2015; Zhang et al., 2017). Interestingly, forming the active complex requires more than just the KASH5 EF-hands and coiled-coil region (amino acids 1–349, Fig. 7). There is a potential spindly motif (Gama et al., 2017) at amino acids 371–376 (Agrawal et al., 2022), which in other adaptors interacts with the pointed end of dynactin (Chaaban and Carter, 2022; Lau et al., 2021). This motif is present in all motility-promoting KASH5 truncations (Fig. 7).

An interesting question is whether dynein's conformation (phi or open) affects its ability to bind adaptors, or subsequently recruit dynactin. If so, then LIS1 binding may regulate adaptor complex assembly per se. While LIS1 does not remain in motile complexes with dynein and dynactin in vitro (Elshenawy et al., 2020; Htet et al., 2020), it appears to be a stable component of at least some DDA complexes in vivo because it is recruited to KASH5 at the NE. It is also found along with dynein and dynactin in KASH5 (Fig. 1 [Horn et al., 2013]) and BICD2 pull-downs (Splinter et al., 2012). It is also required for dynein–dynactin recruitment to the NE in late G2/prophase (Raaijmakers et al., 2013). Understanding fully the in vivo role of LIS1 in dynein–dynactin–adaptor function is a key challenge for the future.

KASH5 interacts with LICs via its N-terminal EF-hand domain, like FIP3, CRACR2a, Rab45, and ninein (Celestino et al., 2019; Lee et al., 2020; Reck-Peterson et al., 2018; Wang et al., 2019). Many EF-hands are known for their regulation by calcium, and CRACR2a-dynein interactions are calcium dependent (Wang et al., 2019). However, not all EF-hand adaptors bind calcium, nor does calcium binding necessarily regulate EF-hand function (Lee et al., 2020; Wang et al., 2019). Indeed, neither KASH5–dynein nor FIP3–dynein interactions was affected by calcium depletion (Fig. S5), and the KASH5 EF-hands lack key residues known to be essential for calcium binding (Fig. 8 A). Furthermore, helix 1 peptide binding to purified KASH5 was not

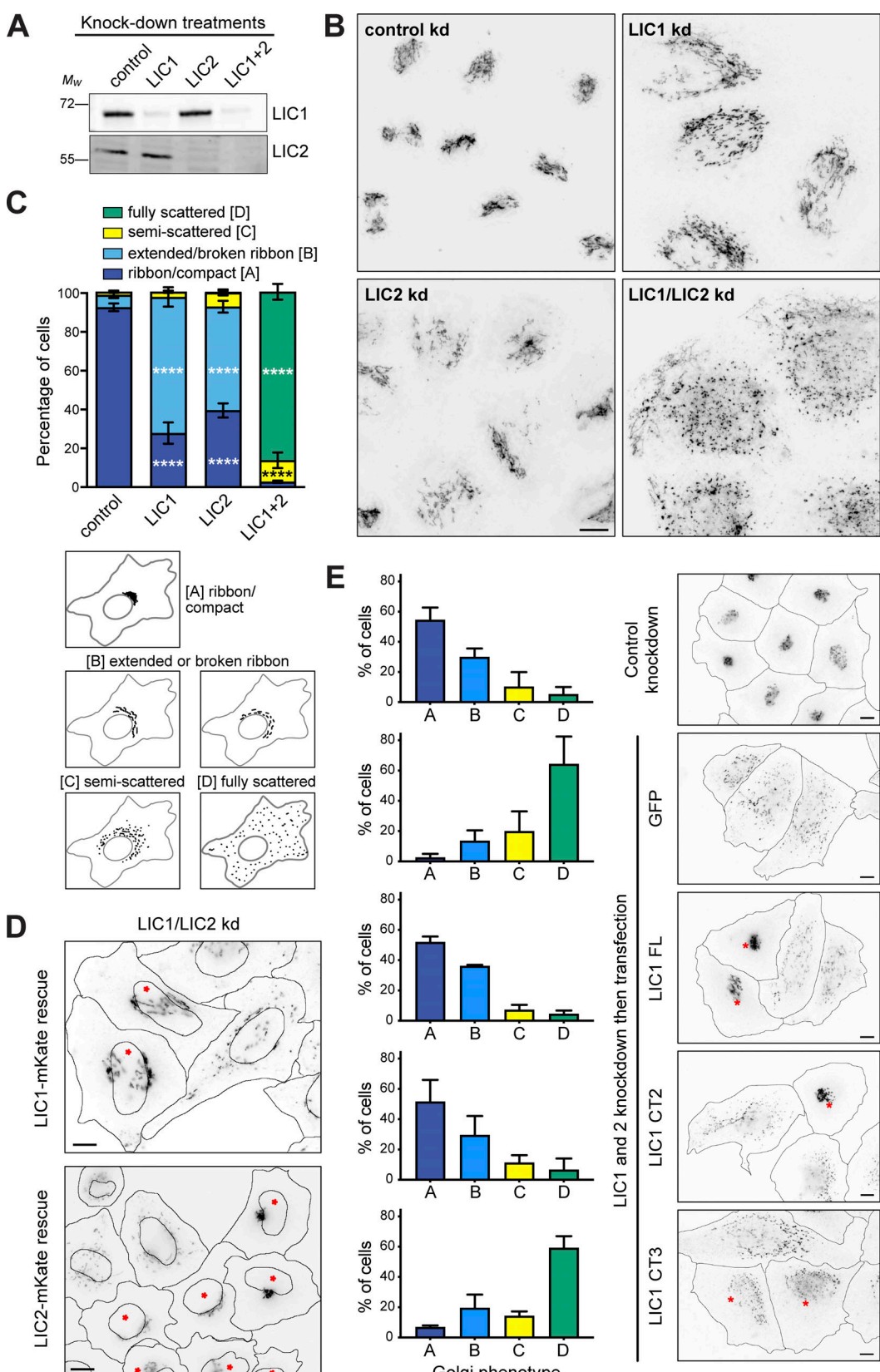

Figure 5. **LICs 1 and 2 act redundantly in Golgi apparatus positioning, with helix 1 being essential. (A–C)** HeLaM cells were depleted of LIC1, LIC2, or both LICs using 5 nM siRNA for each subunit, then analyzed by immunoblotting of lysates with antibodies to LIC1 and 2 (A) or fixed and labeled with antibodies to GM130 and imaged on a DeltaVision microscope to reveal the Golgi apparatus (B). Z-projections of deconvolved images stacks are shown. Quantitation of knock-down efficiency is shown in Fig. S2 C. **(C)** Golgi morphology was scored manually for 100 cells per condition in three independent experiments. Mean ±

SD values are plotted. Statistical analysis was performed using multinomial logistic regression (see Table S2 for full results): comparisons vs. control samples are shown on the graph (P ≤ 0.0001 = ****). **(D and E)** HeLaM cells were depleted of both LICs and then transfected with RNAi-resistant LIC1-mKate or LIC2-mKate (D) or GFP-LIC1-FL, GFP-LIC1-CT2, GFP-LIC1-CT3, and GFP constructs (E). Cells treated with control siRNAs and transfected with GFP were used as controls. GM130 labeling was used to reveal Golgi apparatus morphology (wide-field images: asterisks mark cells expressing the constructs). Golgi morphology was scored for ~100 cells per experimental condition in three independent experiments (E). Mean ± SD values are plotted. GFP and GFP-LIC1-CT3 rescue data are significantly different to control knockdown (P ≤ 0.0001), whereas rescue with GFP-LIC1-FL or GFP-LIC1-CT2 is not (multinomial logistic regression, see Table S2). All scale bars = 10 μm. Source data are available for this figure: SourceData F5.

calcium sensitive and KASH5 did not bind calcium (Agrawal et al., 2022). However, the structure of the KASH5 EF-hand is clearly vital since mutating the X and Y positions of both EF-hands to alanine ablated KASH5–dynein interactions (Fig. 8). Similar adverse effects on dynein–KASH5 association were seen after mutation of other residues in KASH5 EF-hands, and the expression of one of these (L147D) reduced the levels of dynactin somewhat at telomeres in mouse spermatocytes in vivo (Agrawal et al., 2022). Importantly, our studies suggest why the *fue* mutation in the EF-hand (Lindeman and Pelegri, 2012) is detrimental to zebrafish development (Dekens et al., 2003; Lindeman and Pelegri, 2012). It puts a negatively charged glutamic acid in place of a hydrophobic valine in the first exiting helix, which disrupts KASH5–dynein interactions in vitro and in cells (Fig. 8). Interestingly, zebrafish *fue* mutants are defective in pronuclear migration (Dekens et al., 2003). Similarly, Zyg12, the *C. elegans* LINC complex KASH–domain protein that binds dynein LIC via a Hook domain, is needed for both pronuclear migration (Malone et al., 2003; Minn et al., 2009) and meiotic synapsis (Sato et al., 2009). Whether KASH5 plays a role in mammalian pronuclear migration remains to be determined.

As well as characterizing KASH5 as a dynein adaptor, we also provide insight into LIC function in cells. Either LIC can recruit dynein to KASH5 in cells (Fig. 3), as confirmed by using pull-downs with GFP-LIC1 (Figs. 4 and 8) and reconstituting KASH5-activated motility of dynein containing only LIC2 (Fig. 7). Similarly, LICs 1 and 2 act redundantly for recruitment of dynein to RILP, and in the positioning of the Golgi apparatus and recycling, early and late endosomes in cells (Fig. 5 and Fig. S3). We also show that the helix 1-containing region is essential for all these roles. This is in keeping with the roles of Hooks and BicD2 at the Golgi apparatus (Christensen et al., 2021; Hoogenraad et al., 2001; Hoogenraad et al., 2003; Splinter et al., 2012), and Hooks (Christensen et al., 2021; Guo et al., 2016; Olenick et al., 2019; Villari et al., 2020), FIP3 (Horgan et al., 2010a; Horgan et al., 2010b), and RILP (Celestino et al., 2022; Johansson et al., 2007; Jordens et al., 2001; Scherer et al., 2014; Tan et al., 2011) in endocytic organelle dynamics, since all of these adaptors can interact with both LICs (Celestino et al., 2022; Celestino et al., 2019; Christensen et al., 2021; Lee et al., 2020; Lee et al., 2018; Schlager et al., 2014a; Schroeder et al., 2014; Schroeder and Vale, 2016; Urnavicius et al., 2018), although BicD2 may preferentially bind LIC1 in vivo (Goncalves et al., 2019). As yet, however, there is no molecular explanation for why LIC1 is needed for the motility of SNX8-labeled endosomal tubules whereas LIC2-dynein moves SNX1 and SNX4 tubules (Hunt et al., 2013). Other isoform-specific roles for LICs have been described (see Introduction), particularly in mitosis, where LIC phosphorylation, coupled with Pin1 binding, may help control

which adaptors bind to LIC1 and LIC2 (Kumari et al., 2021a; Kumari et al., 2021b). LIC phosphorylation, including in the region just upstream of helix 1, also plays a key role in switching dynein from interphase to mitotic cargos (Addinall et al., 2001; Dwivedi et al., 2019; Kumari et al., 2021a; Niclas et al., 1996).

Altogether, we have shown that KASH5 is a novel transmembrane member of the dynein activating adaptor protein class, mapped its interaction with dynein LICs, and demonstrated that the KASH5 EF-hands are critical for this process. This work also sheds light on the order in which dynein–dynactin–adaptor complexes assemble in cells and the involvement of LIS1 in this process.

## Materials and methods

### Antibodies and constructs

The following mouse antibodies were used: dynein IC (IC74, MAB1618, RRID:AB_2246059; Millipore); EEA1 (Clone 14, RRID:AB_397830; BD Biosciences); LAMP1 (Clone H4A3, RRID: AB_2296838; Developmental Studies Hybridoma Bank; or RRID: AB_470708; Abcam); LIS1 (L7391; RRID:AB_260418; Sigma-Aldrich); dynactin p150 (610473; RRID:AB_397845; BD Transduction Laboratories); GFP (11814460001; RRID:AB_390913; Roche); Myc-tag 9B11 (2276; RRID:AB_331783; Cell Signaling Technology); GM130 (610822; RRID:AB_398141; BD Transduction Laboratories); transferrin receptor (Clone MEM-189, MA1-19300; Thermo Fisher Scientific/Zymed; RRID:AB_2536952); and α-tubulin DM1A (T9026; RRID:AB_477593; Sigma-Aldrich). The following rabbit antibodies were used: HA-tag (H6908; RRID:AB_260070; Sigma-Aldrich); LAMP1 (D2D11, monoclonal; RRID:AB_2687579; Cell Signalling); EEA1 (C45B10, monoclonal; RRID:AB_2096811; Cell Signalling Technology); LIC1 (HPA035013, polyclonal; RRID: AB_10600807; Cambridge Bioscience); LIC2 (ab178702, monoclonal; Abcam); Nde1 (10233-1-AP, polyclonal; RRID:AB_2149877; Proteintech,); p50/dynamitin (EPR5095, ab133492, RRID:AB_11155115; Abcam). Secondary antibodies: LI-COR IRDye secondary antibodies (800CW or 680RD: RRIDs AB_621846, AB_621847, AB_621848, AB_2814912, AB_2716687); Alexa Fluor 488, Alexa Fluor 594, Alexa Fluor 647, Cy3, or Cy5 labeled donkey anti-mouse, -rabbit, and -sheep (RRIDs AB_2340846, AB_2340854, AB_2340813, AB_2340819, AB_2340862, AB_2313584, AB_2340621, AB_2340607, AB_2492288, AB_2307443, AB_2340745, AB_2340748, AB_2340751, AB_2340730; Jackson Immunoresearch).

### Constructs

The following KASH5 constructs in pcDNA 4T/O have been previously described (Horn et al., 2013): GFP-KASH5; GFP-KASH5ΔK; GFP-KASH5ΔNDΔK; GFP-KASH5ΔCDΔK; and GFP-KASH5ND, as has a myc-SUN2 construct in pcDNA3.1(–). EF-hand

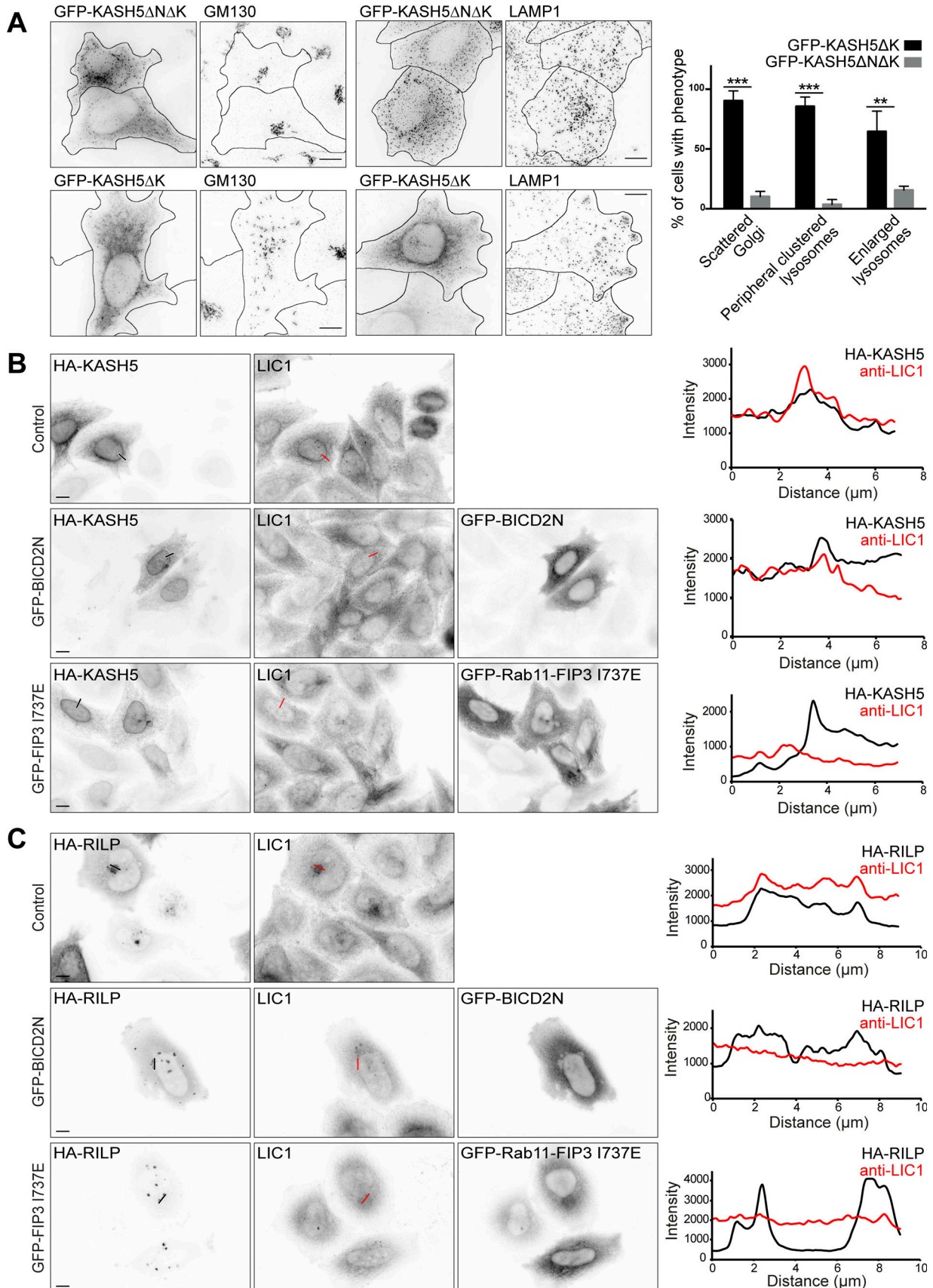

Figure 6. **KASH5 has properties of a dynein adaptor. (A)** HeLaM cells transiently expressing GFP-KASH5ΔK or GFP-KASH5ΔNΔK (which lacks the dynein binding domain) were labeled with GM130 or LAMP1 antibodies (z-stack projections of deconvolved images shown). Cells were scored for phenotypes

associated with dynein inhibition: Golgi apparatus scattering, peripheral clustering of lysosomes, and enlarged lysosomes. Manual scoring of 100 cells per condition was repeated in three independent experiments, with mean and SD shown. An unpaired t test was performed comparing GFP-KASH5ΔK- and GFP-KASH5ΔNΔK-expressing cells for each phenotype. **** = P ≤ 0.0001, *** = P ≤ 0.001, ** = P ≤ 0.01. **(B and C)** Full length HA-KASH5 (B) or HA-RILP (C) were expressed alone (top panel, control) or with dominant negative GFP-BICD2N (middle panel), or dominant negative GFP-Rab11-FIP3-I73E (bottom panel) in HeLaM cells. Endogenous dynein was visualized along with HA-KASH5 or HA-RILP using antibodies to LIC1 and HA (wide-field imaging, scale bar = 10 µm). Thin black and white (left panels) or red lines (LIC1 panels) show where line scan plots were performed (right).

mutants were made using HiFi assembly (New England Biolabs) by linearizing GFP-KASH5 and GFP-KASH5ΔK using Q5 polymerase with primers 5′-GTCATGCGTGACTGGATTGCTG-3′ and 5′-CGTGGAGTTGAGTATTTGCTCCTC-3′ and inserting synthetic G-block sequences (Integrated DNA Technologies) encoding KASH5 bps 97-319 with the appropriate mutations (see Fig. 8). HA-KASH5ΔK was made by amplifying the KASH5ΔK sequence using forward primer 5′-GCGCGGATCCGACCTGCCCGAGGGCCC-3′ and reverse primer 5′-GCGCGAATTCTTATGGATGTCGAGTGACTCTGAGC-3′, digesting with BamH1 and EcoRI, and then ligating into pcDNA3.1 containing the HA tag sequence. For generating a stable cell line, full-length GFP-KASH5 was inserted into the doxycycline-inducible lentiviral vector pTRIPZ (Open Biosystem, Singapore) after amplification using primers 5′-GCGCCTCGAGGACCTGCCCGAGGGCCCGGT-3′ and 5′-GCGCACGCGTTCACACTGGAGGGGGCTGGAGG-3′ followed by digestion with XhoI and MluI. Full-length nesprin2α2 was amplified from a HeLa cDNA library and inserted into pcDNA3.1 downstream of GFP using XhoI and AflII digestion to give GFP-N2α2. A version lacking the transmembrane and KASH domain was generated in the same vector to give GFP-N2α2ΔK.

Silently-mutated siRNA resistant full-length hLIC1-mKate, hLIC2-mKate, and hLIC2-GFP have been previously described (Jones et al., 2014). RNAi-resistant full-length LIC1 and LIC2 and LIC1 truncations were generated by PCR and restriction digest cloning into pEGFP-C3 (Clontech) and used for rescue experiments. GFP-LIC1-CT2 encodes amino acids 1–456 of human LIC1 with the addition of the amino acid sequence ADPPDLDN after the LIC1 C-terminus. GFP-LIC1-CT3 encodes amino acids 1–387 followed by ADPPDLDN. N-terminally Myc-tagged versions of full-length human LIC1, LIC1-CT2, and LIC1-CT3 were generated using the forward primer 5′-CAGCTGGTACCGCGGCCGTGGGGCGAGTC-3′. The reverse primers were 5′-TCGAATCTAGACTAAGAAGCTTCTCCTTCCGTAGGAGATG-3′ (full length LIC1), 5′-TCGAATCTAGACTAAGAGCCAGTCTTTTTTACTCAACAAAC-3′ (LIC1-CT2), and TCGAATCTAGACTATGGTGGTTGCTTTGCTAAAAGGGAC (LIC1-CT3), with no additional C-terminal amino acids. PCR products were inserted into pcDNA-3.1 downstream of a myc-tag sequence using KpnI and XbaI digestion. To generate LIC1 helix 1 mutants (FF: F447A, F448A and AA: L451A, L452A), EGFP-LIC1 was linearized using Q5 polymerase (New England Biolabs) with primers 5′-GTGTGAGTGGTGGTAGCCCTG-3′ and 5′-TGTAGCTCCAGCTTTCATGTTTGG-3′. A 111 bp single-stranded ultramer (IDT) was used to insert the appropriate mutated sequence using HiFi assembly (New England Biolabs). LIC1 helix 1 deletion mutants were generated in EGFP-LIC1 by Q5 amplification with phosphorylated primers followed by blunt-end ligation using the NEB Quick Ligation kit. The primers for deletion of helix 1 (S440-T456) were 5′-Phos-

GGCTCTCCAGGAGGCCCTG-3′ and 5′Phos-TGTAGCTCCAGCTTTCATGTTTGGATC-3′.For deletion of helix 1 and flanking sequences (N433-S458), the primers were 5′Phos-CCAGGAGGCCCTGGTGTGAG-3′ and 5′Phos-TGGATCAATTTTTTTTGACCCAGCAGGAA-3′.

Vectors encoding mCherry-chicken p50 and RFP-CC1 have been previously described (Wozniak et al., 2009). To generate GFP-BICD2N, the DNA sequence encoding amino acids 2–402 of mouse BICD2 was amplified using forward primer: 5′-GCGCGAATTCGTCGGCGCCGTCGGAGGAG-3′ and reverse primer: 5′-GCGCGGATCCTCACAGGCGCCGCAGGGCACT-3′, and then cloned into pEGFP-C1 (Clontech) using EcoRI and BamHI. Other constructs were generous gifts from the following colleagues: pMDG2.1-VSV-G and p8.91-Gag-Pol vectors (Apolonia et al., 2015; Zufferey et al., 1997), Dr. M. Malim (King's College London, UK); GFP-Rab11-FIP3 and GFP-Rab11-FIP3 I737E, (Wilson et al., 2005), Prof. G. Gould (University of Glasgow, UK); pEGFP-C1 containing hRILP (Colucci et al., 2005), Prof. Cecilia Bucci (University of Salento, Italy); and pCB6-HA-RILP, Dr. Mark Dodding (University of Bristol, UK).

## Cell lines and transfection

HeLa and hTERT-RPE cells were obtained from ATCC; Vero cells were purchased from the European Collection of Authenticated Cell Cultures. HeLaM cells were kindly provided by Dr. Andrew Peden, University of Sheffield, UK. Mycoplasma testing was routinely performed by DAPI staining.

HeLa, HeLaM, HEK293T, and Vero cells were maintained in DMEM supplemented with 10% FBS at 8.0% $CO_2$ and 37°C. To generate cells stably expressing human GFP-KASH5, HeLa cells were transfected with pTRIPZ-GFP-KASH5, pMDG2.1-VSV-G, and p8.91-GAG-POL in a ratio of 4:1:2 to a total of 10 µg of DNA using Lipofectamine 2000 (Thermo Fisher Scientific) as per the manufacturer's instructions overnight. The virus was then collected and passed through a 0.45-µm filter before addition to cells for 4 h. Transduced cells were selected with fresh media containing puromycin (3 µg/ml). Induction of GFP-KASH5 expression was induced with doxycycline (500 ng/ml) for 16 h. Transient transfection of HeLaM and Vero cells on #1.5 coverslips was achieved using JetPEI (PolyPlus transfection) using half the manufacturer's recommended amounts of total DNA and JetPEI. Expression levels were carefully titrated for each plasmid, in some cases using dilution with carrier DNA (pBluescript SK-II; Flores-Rodriguez et al., 2011), to avoid over-expression artifacts. For biochemical analysis, cells were transfected in 10-cm dishes using either JetPEI, PEI (408727; Sigma-Aldrich) or FuGENE HD (E2311). PEI was dissolved at 1 mg/ml in 150 mM NaCl by incubation at 50°C, sterile-filtered, and stored in aliquots at −80°C. Per dish, 16 µg of total DNA was diluted in 200 µl

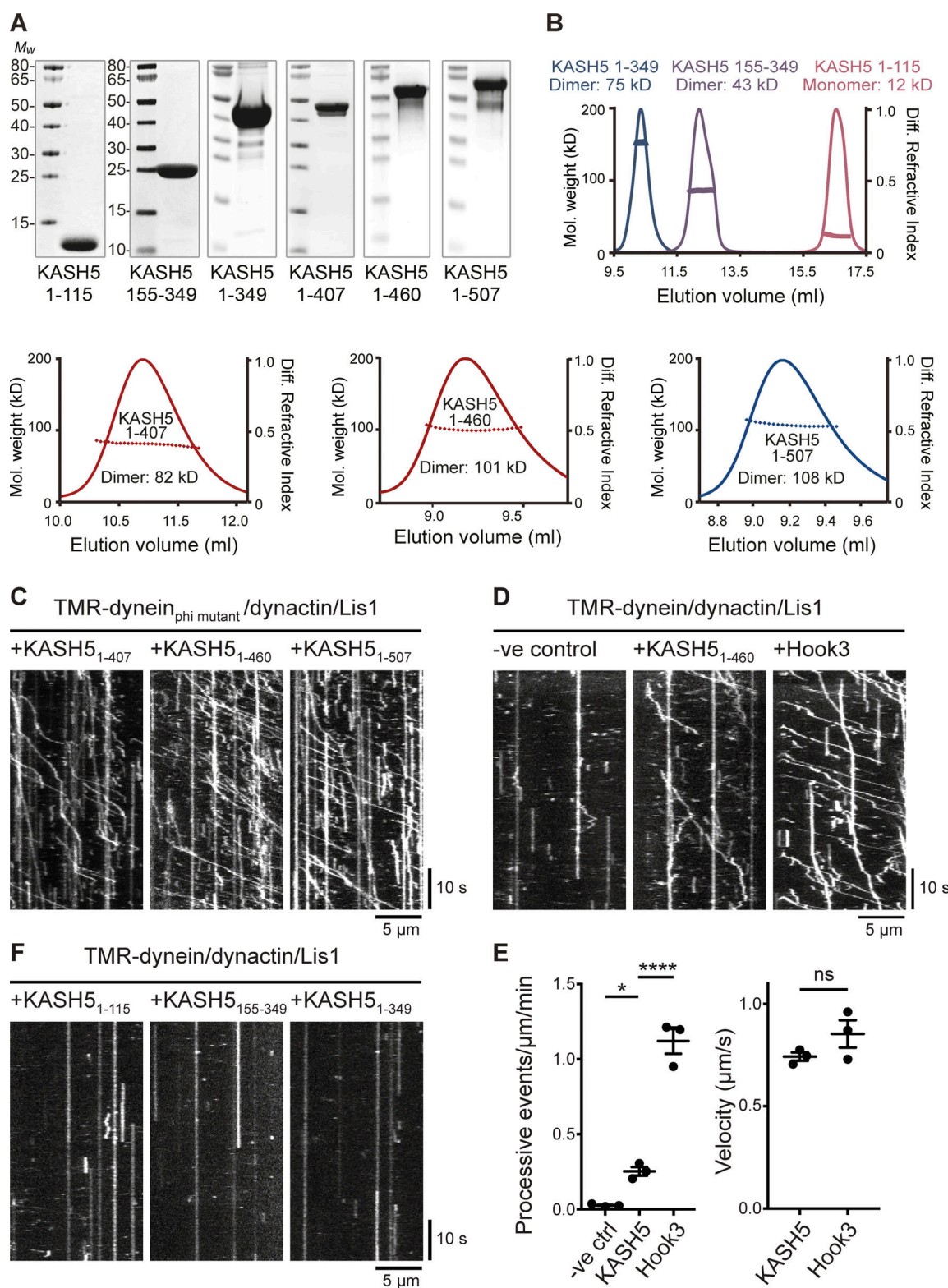

Figure 7. **KASH5 is an activating adaptor for dynein motility in vitro. (A)** SDS-PAGE and Coomassie blue staining of purified bacterially expressed KASH5 truncations. KASH5$_{1-115}$ contains the EF hands; KASH5$_{155-349}$ consists of only the coiled coil; KASH5$_{1-349}$ is the EF-hand plus coiled coil; KASH5$_{1-407}$, KASH5$_{1-460}$, and KASH5$_{1-507}$ all contain the EF hands, coiled coil, and additional sequence. **(B)** SEC-MALS analysis of purified KASH5 truncations, showing the measured $M_w$. The predicted dimeric $M_w$ values are: KASH5$_{1-407}$, 92 kDa; KASH5$_{1-460}$, 103 kDa; and KASH5$_{1-507}$, 114 kDa. **(C)** Purified baculovirus-expressed recombinant dynein containing a heavy chain R1567E/K1610E mutant that cannot form the inhibited Phi conformation, LIS1, and porcine brain dynactin were combined with KASH5$_{1-407}$, KASH5$_{1-460}$, or KASH5$_{1-507}$ and motility of individual 6-carboxytetramethylrhodamine-labeled dynein molecules along microtubules was visualized using TIRF microscopy and displayed as kymographs. **(D)** Motility of wild-type recombinant dynein, LIS1, and dynactin in the presence or absence of KASH5$_{1-460}$, with the activating adaptor Hook3$_{1-522}$ as a positive control. **(E)** Analysis of motility from D. Left panel: the number of processive events per μm microtubule

per minute was determined from kymographs in a blinded fashion for all three conditions in three technical replicates, with the mean ± SD plotted. The total number of movements analyzed were 2066 for Hook3, 339 for KASH5, and 34 for the no additional control. Significance was determined using ANOVA with Tukey's multiple comparison (ns = not significant, * = P ≤ 0.05, **** = P ≤ 0.0001). Right panel: the mean velocity of processive dynein movements from the KASH5$_{1-406}$ and Hook3$_{1-522}$ data are plotted (± SD, $n$ = 3 replicates). **(F)** Kymographs of 6-carboxytetramethylrhodamine-labeled wild-type dynein molecules combined with LIS1, dynactin, and KASH5$_{1-115}$, KASH5$_{155-349}$, or KASH5$_{1-349}$. Source data are available for this figure: SourceData F7.

Opti-MEM (319850; Gibco), 48 µl of PEI was added to another 200 µl of Opti-MEM, and then after 5 min the PEI was added to the DNA mix and incubated for 30 min at room temperature before adding to the cells.

## Short-interfering RNA (siRNA) methods

For depletion of target genes, siRNA transfections used INTER-FERin (PolyPlus Transfection). siRNAs targeting the ORF of human LIC1 and LIC2 were obtained from Eurofins MWG Operon. Oligonucleotides were applied to HeLaM cells at a final concentration of 5–20 nM (for HeLaM) or 20 nM (for GFP-KASH5 HeLas) per target for 72 h before analysis by immunoblot and immunofluorescence. The following sequences were used, synthesized by Eurofins MWG with dTdT overhangs: LIC1, 5′-AGAUGACAGUGUAGUUGUA-3′; LIC2, 5′-ACCUCGACUUGU UGUAUAA-3′ (Jones et al., 2014; Palmer et al., 2009); and LIS1, 5′-GAACAAGCGAUGCAUGAAG-3′ (Lam et al., 2010; Tsai et al., 2005). For IC2, a SMARTpool (Thermo Fisher Scientific Dharmacon) was used consisting of a mixture of four siRNAs: 5′-GUA AAGCUUUGGACAACUA-3′; 5′-GAUGUUAUGUGGUCACCUA-3′; 5′-GCAUUUCUGUGGAGGGUAA-3′; and 5′-GUGGUUAGUUGU UUGGAUU-3′. Control RNAi experiments were performed either using siGENOME lamin A/C Control (Thermo Fisher Scientific: Fig. 5, B and C; and Fig. S3, A and B) or ON-TARGET-plus Non-targeting siRNA #1 (Thermo Fisher Scientific Dharmacon). For transfection of siRNA-treated cells for biochemical or immunofluorescence analysis, cells were depleted for 24 or 48 h and then transfected with either JetPEI, PEI, or FuGENE HD and incubated for a further 24 h before lysate preparation or fixation. For LIC rescue experiments, the LIC1 constructs used were siRNA resistant (Jones et al., 2014).

## Immunoblotting

For validating siRNA knock-down efficiency, lysates were collected 72 h post siRNA addition in RIPA lysis buffer (R0278; Sigma-Aldrich) supplemented with cOmplete ULTRA protease inhibitor (5892791001; Roche) and PhosSTOP (Roche PHOSS-RO) phosphatase inhibitor. A total of 20 µg of protein was loaded per well diluted in denaturing SDS loading buffer, and protein samples and molecular weight markers (PageRuler Plus Pre-stained Protein Ladder, Thermo Fisher Scientific 26619, or Colour pre-stained protein standards NEB P7712 or P7719) were separated using 8, 10, or 12% polyacrylamide gels before being transferred to PVDF Immobilon-FL membrane with pore size 0.45 µm (Millipore Billerica) using a Mini Trans-Blot electrophoretic transfer system (BioRad). The transfers were carried out in an ice-cold transfer buffer at 100 V for 1.5 h. After transfer, the membrane was blocked either by incubation in 5% dried milk (Marvel) in PBS with 0.1% Tween-20 (Sigma-Aldrich; PBST) for 30 min at room temperature or in 1× alpha-casein

buffer (B6429; Sigma-Aldrich) diluted in tris-buffered saline (TBS: 20 mM Tris/HCl, pH 7.7, 150 mM NaCl) for 1 h at room temperature. Primary antibodies were diluted in 5% milk/PBST or 1× alpha-casein diluted in TBS supplemented with 0.01% Tween-20 (TBST) and incubated at 4°C overnight. Membranes were washed three times in TBST or PBST and incubated with LI-COR IRDye secondary antibodies (800CW or 680RD) diluted 1:10,000 in 5% milk/PBST or 1× alpha-casein in TBST for 1 h at room temperature. Blots were washed three times in PBST or TBST and once in water before imaging on a LI-COR Odyssey or Odyssey CLx using Image Studio software.

## GFP-Trap immunoprecipitation

For LIC1 truncation or mutant immunoprecipitation experiments, HeLa M cells were depleted of endogenous LIC1 and LIC2 for 24 h prior to transfection. Cells were transiently transfected using PEI or JetPEI (see Cell lines and transfection methods) and incubated for 24 h. Cells were washed with ice-cold PBS and lysed in IP lysis buffer (50 mM Tris, pH 7.5, 10 mM NaCl, 2.5 mM MgCl$_2$, 1 mM DTT, 0.01% digitonin [300410; Calbiochem] and protease inhibitors, 10 µg/ml aprotinin [A6103; Sigma-Aldrich], leupeptin [108976; Calbiochem], and pepstatin [J60237; Alfa Aesar]) before centrifugation at 17,000 rpm for 30 mins at 4°C in a microcentrifuge. For EGTA treatment, 1 mM EGTA was added to the lysis buffer at this point. A tenth of the supernatant was taken as an input sample and the remaining supernatant was rotated at 4°C for 2 h with ChromoTek GFP-Trap magnetic agarose beads prewashed in IP wash buffer (50 mM Tris, pH 7.5, 10 mM NaCl, 1 mM DTT, and protease inhibitors). For EGTA treatment, 1 mM EGTA was added to the IP wash buffer. Beads were isolated and washed three times in IP wash buffer, and proteins were eluted by boiling at 95°C in SDS-PAGE sample buffer before analysis by SDS-PAGE and immunoblotting. For LIC mutant immunoprecipitations, cells were transfected using FuGENE HD transfection reagent, as per manufacturer's instructions, with a reagent:DNA ratio of 3:1 and incubated for a further 24 h following endogenous LIC depletion. The buffers contained 50 mM NaCl and 0.01% digitonin was added to the wash buffer.

## BAPTA-AM treatment

HeLaM cells were grown on uncoated #1.5 glass coverslips and Vero cells were grown on coverslips coated with 1 µg/ml Fibronectin (F0895; Sigma-Aldrich) and then transfected. Cells were treated with either 10 µM BAPTA-AM (ab120503; Abcam) or DMSO vehicle control for 2 h before fixation and immunofluorescence labeling.

## Quantification of immunoblots

Protein band signal intensity was measured using LI-COR Image Studio Lite software or ImageJ and blot-background signal

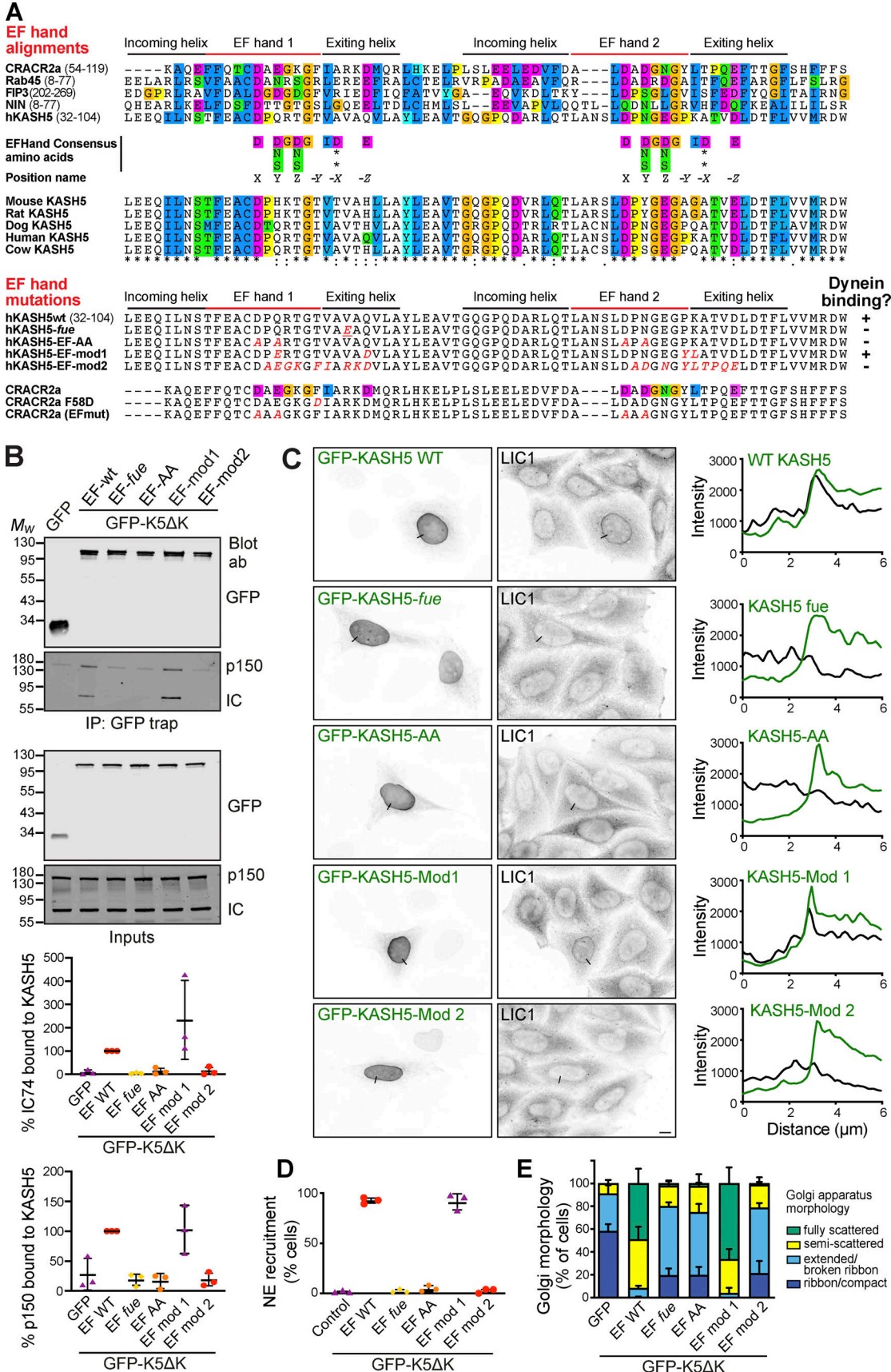

Figure 8. **KASH5's EF-hand is critical for dynein and dynactin complex assembly. (A)** Top: Sequence comparison between human KASH5 and other dynein adaptors containing EF-hands, and between KASH5 proteins from different species. The EF-hand consensus sequence is shown, along with position

nomenclature (Grabarek, 2006). Bottom: Mutations generated in KASH5 EF-hands (altered amino acids shown in red), along with published mutations in the calcium-dependent dynein adaptor CRACR2a (Wang et al., 2019). Summary of KASH5 mutant dynein-binding activity, as indicated by + or − symbols. **(B)** GFP-trap immunoprecipitates from cells expressing GFP-hKASH5ΔK, GFP-hKASH5ΔK EF hand mutants, or GFP, probed with antibodies to GFP, dynein IC, and dynactin p150. Quantitation of IC and p150 levels normalized to GFP-LIC1-WT levels (n = 3 experiments) is plotted graphically below. **(C and D)** HeLaM cells were transiently transfected with GFP-KASH5-WT or EF hand mutants along with HA-SUN1 (not shown). Cells were fixed and labeled with antibodies against endogenous dynein LIC1 and GFP. **(C)** Line scans (black lines on the images) were performed and are shown on the right: LIC1 shown in black, GFP-KASH5 in green. Wide-field imaging; scale bar = 10 µm. **(D)** The samples were scored for dynein recruitment (see Materials and methods; 100 cells per condition, in each of three independent repeats). **(E)** GFP-tagged KASH5ΔK, GFP-KASHΔK EF-hand mutants, or GFP were expressed in HeLaM cells then fixed and labeled with antibodies to GFP and GM130 then scored for Golgi apparatus morphology in a blinded fashion (100 cells per condition, in each of three independent repeats). All graphs show mean values ± SD. Statistical analysis is given in Table S3. Source data are available for this figure: SourceData F8.

---

intensity was subtracted. For siRNA knock-down experiments, band signal intensity was normalized to total protein. The protein amount was then expressed as a percentage compared to control siRNA samples, which were set as 100%. For immunoprecipitation experiments, the enrichment factor for GFP or each GFP-tagged protein in the pull-down was first calculated as bound/input = $x$. For each prey, bound/input = $y$ was calculated and then the efficiency of pull-down compared to GFP was determined as a percentage ($y/x$*100). For GFP-trap pull-downs from siRNA-treated cells, the prey protein levels could not be normalized compared to input levels if that protein had been depleted. Instead, the amount of each prey protein in the control-depleted GFP-KASH5DK pull-downs was set as 100% and the amount present in other samples was compared to that.

### Immunofluorescence

HeLaM or Vero cells were grown on #1.5 coverslips at an appropriate density. For EEA1 antibody labeling, cells were fixed for 20 mins in 3% formaldehyde in PBS at RT. After washing, unreacted formaldehyde was quenched with glycine (0.1 M) and permeabilized in 0.1% Triton X-100 in PBS for 10 min before PBS wash, prior to antibody labeling. For all other antibody labeling, cells were fixed in methanol for 10 min at −20°C and washed in PBS. Secondary antibodies labeled with Alexa Fluor 488, Alexa Fluor 594, Alexa Fluor 647, Cy3, or Cy5 were used along with 1 µg/ml DAPI, and samples were mounted in ProLong Gold (Invitrogen). For LIS1 antibody labeling (Baffet et al., 2015), coverslips were washed once in PBS and then incubated for 1 min in PHEM buffer (60 mM PIPES, 25 mM HEPES, 10 mM EGTA, 2 mM MgCl$_2$, pH 6.9) containing 0.5% Triton X-100. Cells were then transferred to PHEM buffer supplemented with 3.7% paraformaldehyde for 20 min. This was followed by incubation in 0.2% Triton X-100 in PBS for 5 min. Coverslips were rinsed in PBS and quenched with glycine before labeling.

### Imaging and analysis

Fixed HeLaM (Fig. 4 D; Fig. 5, D and E; Fig. 6, B and C; Fig. 8 C; Fig. S3, C and D; Fig. S4 E; and Fig. S5 A) and Vero cells (Fig. S5 B) were imaged using an Olympus BX60 or BX50 microscope (Olympus Keymed) with a Plan apochromat 60X 1.4 N.A. oil immersion objective, CoolLED light source (CoolLED Ltd) or 100W Hg lamp, CoolSNAP ES CCD camera (Teledyne Photometrics) and MetaVue software (Molecular Devices). Subsequent image analysis was performed using MetaVue and ImageJ software. The imaging in Figs. 1, 2, and 3; Fig. 5 A, Fig. 6 A, and Fig. S3 A was performed on a IX71 microscope (Olympus), which

was equipped for optical sectioning by a DeltaVision CORE system (GE Healthcare) with z-spacing fixed at 0.2 µm. Plan Apochromat 60X 1.4 N.A. or UPlanFl 100X 1.35 N.A. oil immersion objective lenses (Olympus) were used with a CoolSNAP HQ CCD camera (Photometrics). Deconvolution was completed using SoftWorX (Applied Precision) and images are displayed as z-projections either using SoftWorX or using the ImageJ extended depth of field plugin.

Line scans were generated using ImageJ using the plot profile tool, with the lines starting in the cytoplasm. Data was exported and plotted as graphs in GraphPad Prism. ImageJ was used to generate inverted greyscale images, merge channels, and draw cell outlines, using contrast-adjusted images as a guide, where needed. Some images were prepared and annotated using Adobe Photoshop. For phenotype scoring, each experiment was performed at least three times and the number of cells manually scored is given in the figure legends. Examples of the Golgi apparatus, early endosome, and lysosome phenotypes are shown in Fig 5 and Fig. S3. For dynein recruitment to KASH5 in Fig. 3 D, Fig. 4 E, Fig. 8 C, and Fig. S1, C and D, cells were scored in a binary fashion as to whether dynein or dynactin could be seen at the NE. For the analysis shown in Figs. 4 and 8, the scoring was performed without knowing the sample identity. Cells with strong nuclear enrichment of GFP-KASH5 and HA-SUN1 (Fig. 8) or GFP-KASH5 and myc-SUN2 (Fig. 4), but with limited accumulation in the ER, were selected without reference to the LIC1 channel, which was then used to score for dynein recruitment. For these data, statistical tests were not deemed appropriate as there was no variation in some control conditions, with all cells recruiting dynein or dynactin to KASH5. For other datasets, the statistical tests used are given in the figure legends. For experiments where one condition was compared with a control, an unpaired t-test was performed using GraphPad Prism (GraphPad Software). For multiple comparisons, one-way ANOVA was used if all data points were present. In Fig. 4 E, ANOVA could not be used because four data points were missing (out of a possible 49). Instead, a mixed model was implemented using Prism, fitted using Restricted Maximum Likelihood (REML) with Geisser-Greenhouse correction. Tukey's multiple comparison was used for ANOVA and REML analysis. For experiments where categoric phenotype scoring data across multiple conditions were compared to one control, multinomial logistic regression was applied using SPSS (IBM) software (Lam et al., 2010). Full statistical analysis is given in Tables S2, S3, and S4. Graph preparation was performed in Prism. Figures were assembled using Adobe Illustrator (Adobe) or Affinity Designer (Serif Europe Ltd).

## Recombinant KASH5 protein expression and purification

Constructs of human KASH5 (amino acid residues: 1-115, 155-349, 1-349, 1-407, 1-460, and 1-507) were cloned into pMAT11 vector (Peranen et al., 1996) for expression with an N-terminal TEV-cleavable His$_6$-MBP tag. The primers used were: KASH5_1_forward primer: 5′-TTCCAGGGTTCCATGGACCTGCCTGAGGGCCCG GTGGGTGGC-3′; KASH5_115_reverse primer: 5′-AATTCGATA TCCATGGTTATTATTCCAGACCACCATGCAGCTGACATGCTGC-3′; KASH5_155_forward primer: 5′-TTCCAGGGTTCCATGGAA CTGCAGGCAACAGCCGAT-3′; KASH5_349_reverse primer: 5′-AATTCGATATCCATGGTTATTATTCATAGGTCTGGCTCAGCTG-3′; KASH5_407_reverse primer: 5′-AATTCGATATCCATGGTTATT ATTCGTGAATAACTTCTTC-3′; KASH5_460_reverse primer: 5′-AATTCGATATCCATGGTTATTAATCAGCCGTGACCTGGCTCTCTG C-3′; KASH5_507_reverse primer: 5′-AATTCGATATCCATGGTT ATTACTGGCCCCAGGCCCTCCTCCTGACTGG-3′. KASH5 constructs were expressed in BL21 (DE3) cells (Novagen), in 2xYT media, and induced with 0.5 mM IPTG for 16 h at 25°C. Bacterial pellets were harvested, resuspended in 20 mM Tris, pH 8.0, 500 mM KCl, and lysed using a TS Cell Disruptor (Constant Systems) at 172 MPa. Cellular debris was later removed by centrifugation at 40,000 $g$. KASH5 fusion proteins were purified through consecutive Ni-NTA (Qiagen), amylose (NEB), and HiTrap Q HP (Cytiva) ion exchange chromatography. The N-terminal His$_6$-MBP tag was cleaved using TEV protease, and the cleaved samples were further purified through HiTrap Q HP (Cytiva) ion exchange chromatography and size exclusion chromatography (HiLoad 16/600 Superdex 200, Cytiva) in 20 mM HEPES, pH 7.5, 150 mM KCl, 2 mM DTT. Purified KASH5 protein samples were spin-concentrated using Amicon Ultra centrifugal filter device (10,000 NMWL), flash-frozen in liquid nitrogen, and stored at –80°C. Purified KASH5 proteins were analyzed using SDS-PAGE and visualized with Coomassie staining. Protein concentrations were determined using Cary 60 UV spectrophotometer (Agilent) with extinction coefficients and molecular weights calculated by ProtParam (http://web.expasy.org/protparam/).

## Multiangle light scattering coupled with size exclusion chromatography (SEC-MALS)

The absolute molar masses of KASH5 protein samples were determined by multiangle light scattering coupled with size exclusion chromatography (SEC-MALS). KASH5 protein samples at >1.5 mg ml$^{-1}$ were loaded onto a Superdex 200 Increase 10/300 GL size exclusion chromatography column (Cytiva) in 20 mM Tris, pH 8.0, 150 mM KCl, 2 mM DTT, at 0.5 ml min$^{-1}$, in line with a DAWN HELEOS II MALS detector (Wyatt Technology) and an Optilab T-rEX differential refractometer (Wyatt Technology). Differential refractive index and light scattering data were collected and analyzed using ASTRA 6 software (Wyatt Technology). Molecular weights and estimated errors were calculated across eluted peaks by extrapolation from Zimm plots using a dn/dc value of 0.1850 ml g$^{-1}$. Bovine serum albumin (Thermo Fisher Scientific) was used as the calibration standard.

## Protein purification

Human LIS1 (Baumbach et al., 2017) and full-length wild type human cytoplasmic dynein-1 (Schlager et al., 2014a), or a version containing heavy chain phi-disrupting mutations R1567E and K1610E that prevent the phi autoinhibitory conformation of dynein (Zhang et al., 2017), were expressed using the *Sf*9/baculovirus system and purified as previously described (Baumbach et al., 2017; Schlager et al., 2014a). Pellets from 1 liter of Sf9 cell culture were suspended in 50 ml lysis buffer (50 mM HEPES, pH 7.2, 100 mM NaCl, 1 mM DTT, 0.1 mM Mg.ATP, and 10% Glycerol), supplemented with one cOmplete tablet (Roche) and 1 mM PMSF. Cells were lysed using a Dounce homogenizer at 4°C, then lysate-clarified for 45 min at 500,000 × $g$. The supernatant was incubated with 1.5 ml pre-equilibrated IgG beads (Cytiva) for 3 h. Beads were washed with 200 ml lysis buffer. For dynein, beads were then transferred to a 2 ml tube, adding 10 µM SNAP-Cell TMR-Star dye (New England Biolabs) and incubating for 1 h at 4°C. Beads for both constructs were then washed with 100 ml TEV buffer (50 mM Tris-HCl, pH 7.4, 148 mM K-acetate, 2 mM Mg-acetate, 1 mM EGTA, 10% Glycerol, 0.1 mM Mg.ATP, and 1 mM DTT) and then transferred again to a 2 ml tube, adding 400 µg TEV protease and incubating overnight at 4°C. The sample was then concentrated and gel filtered using a G4000$_{SWXL}$ 7.8/300 column (TOSOH Bioscience) into GF150 buffer (25 mM HEPES, pH 7.2, 150 mM KCl, 1 mM MgCl$_2$, 5 mM DTT, 0.1 mM Mg.ATP) for dynein or TEV buffer for Lis1. Peak fractions were concentrated and snap-frozen in liquid nitrogen.

Dynactin was purified from frozen porcine brains as previously described (Urnavicius et al., 2015). Porcine brains were blended with 300 ml homogenization buffer (35 mM PIPES, pH 7.2, 5 mM MgSO$_4$, 1 M KCl, 200 µM EGTA, 100 µM EDTA, 1 mM DTT) supplemented with four cOmplete tablets and 1 mM PMSF until just thawed. The lysate was then clarified in two steps: 15 min at 38,400 × $g$ at 4°C, then 50 mins at 235,000 × $g$ at 4°C. All subsequent steps were carried out at 4°C. The supernatant was then filtered through a GF filter then a 0.45 µm filter, then loaded onto an SP Sepharose column (Cytiva), equilibrated with SP buffer A (35 mM PIPES, pH 7.2, 5 mM MgSO$_4$, 1 mM EGTA, 0.5 mM EDTA, 1 mM DTT, and 0.1 mM Mg.ATP). The resin was then washed until white using 99.5% SP buffer A/0.5% SP buffer B (35 mM PIPES, pH 7.2, 5 mM MgSO$_4$, 1 M KCl, 1 mM EGTA, 0.5 mM EDTA, 1 mM DTT, and 0.1 mM Mg.ATP), and then dynactin was eluted using a gradient from 0.5 to 25% (% SP buffer B). Dynactin-containing fractions were then filtered through a 0.22-µm filter, then loaded onto a Mono Q HR 16/60 column (Cytiva), pre-equilibrated with Mono Q buffer A (35 mM PIPES, pH 7.2, 5 mM MgSO$_4$, 200 µM EGTA, 100 µM EDTA, 1 mM DTT). This column was washed with 10 CV Mono Q buffer A and then dynactin-eluted using a 15–35% gradient (% Mono Q buffer B, 35 mM PIPES, pH 7.2, 5 mM MgSO$_4$, 1 M KCl, 200 µM EGTA, 100 µM EDTA, and 1 mM DTT). Peak fractions were concentrated and gel-filtered using a G4000$_{SW}$ 21.5/600 column (TOSOH Bioscience) into GF150 buffer, with the peak concentrated and snap-frozen in liquid nitrogen.

Strep-tagged human Hook3 (1-522; Urnavicius et al., 2018) was purified using the *Sf*9/baculovirus system as previously described (Urnavicius et al., 2018). A pellet from 500 ml of Sf9 culture was thawed using 50 ml Strep-tag lysis buffer (30 mM HEPES, pH 7.2, 50 mM K-Acetate, 2 mM Mg-Acetate, 1 mM EGTA, 10% Glycerol, and 1 mM DTT) plus one cOmplete tablet

and 1 mM PMSF. Cells were then lysed using a Dounce homogenizer, with the lysate clarified for 20 min at 50,000 × g at 4°C. The supernatant was filtered using a GF filter, then flown onto a StrepTrap HP 1 ml column (Cytiva) at 4°C, washed using 40 CV Strep-tag lysis buffer, and then eluted with 3 mM desthiobiotin. Peak fractions were concentrated and gel-filtered using a Superose 6 10/300 Increase column (Cytiva) into GF150 buffer. The monodisperse peak was concentrated and snap-frozen in liquid nitrogen.

**In vitro TIRF motility assays**
In vitro TIRF assays were carried out as previously described (Urnavicius et al., 2018). Microtubules were typically prepared the day before the assay. Microtubules were made by mixing 1 µl of HiLyte Fluor 488 tubulin (2 mg/ml, Cytoskeleton), 2 µl biotinylated tubulin (2 mg/ml, Cytoskeleton). and 7 µl unlabelled pig tubulin (Schlager et al., 2014a; 6 mg/ml) in BRB80 buffer (80 mM PIPES, pH 6.8, 1 mM $MgCl_2$, 1 mM EGTA, 1 mM DTT). 10 µl of polymerization buffer (2× BRB80 buffer, 20% (v/v) DMSO, 2 mM Mg.GTP) was added and then the solution was incubated at 37°C for 1 h for microtubule polymerization. The sample was diluted with 100 µl of MT buffer (BRB80 supplemented with 40 µM paclitaxel) and then centrifuged at 21,000 × g for 9 min at room temperature to remove soluble tubulin. The resulting pellet was gently resuspended in 100 µl of MT buffer and then centrifuged again as above. 50 µl MT buffer was then added, with the microtubule solution then stored in a light-proof container. Before usage, and every 5 h during data collection, the microtubule solution was spun again at 21,000 × g for 9 min, with the pellet resuspended in the equivalent amount of MT buffer.

Assay chambers were prepared by applying two strips of double-sided tape on a glass slide, creating a channel, then placing a piranha-solution-cleaned coverslip on top. The coverslip was then functionalized using PLL-PEG-Biotin (SuSOS), washed with 50 µl of TIRF buffer (30 mM HEPES, pH 7.2, 5 $MgSO_4$, 1 mM EGTA, 2 mM DTT), and then incubated with streptavidin (1 mg/ml; New England Biolabs). The chamber was again washed with TIRF buffer and then incubated with 10 µl of a fresh dilution of microtubules (2 µl of microtubules diluted into 10 µl TIRF-Casein buffer [TIRF buffer supplemented with 50 mM KCl and 1 mg/ml casein]) for 1 min. Chambers were then blocked with a 50 µl blocking buffer.

Complexes were prepared by mixing 1.5 µl of each component at the following concentrations: dynein at 0.3 µM, dynactin at 0.3 µM, adaptor at 6 µM, and Lis1 at 50 µM. GF150 buffer was added to a final volume of 6 µl. Complexes were incubated on ice for 15 min and then diluted with TIRF-Casein buffer to a final buffer of 15 µl. Four microliters of the complex were added to 16 µl of TIRF-Casein buffer supplemented with an oxygen scavenging system (0.2 mg/ml catalase, Merck; 1.5 mg/ml glucose oxidase, Merck; 0.45% [w/v] glucose) 1% BME, and 5 mM Mg.ATP. This mix was flowed into the chamber.

The sample was imaged immediately at 23°C using a TIRF microscope (Nikon Eclipse Ti inverted microscope equipped with a Nikon 100× TIRF oil immersion objective). For each sample, a microtubule image was acquired using a 488 nm laser. Following this, a 500-frame movie was acquired (200 ms exposure, 4.1 fps) using a 561 nm laser. To analyze the data, ImageJ was used to generate kymographs from the tiff movie stacks. These kymographs were blinded and then events of similar length were picked to analyze velocity and the number of processive events/µm microtubule/s, using criteria outlined previously (Schlager et al., 2014a; Urnavicius et al., 2018). Velocity was calculated using a pixel size of 105 nm and a frame rate of 235 ms/frame. Three replicates were taken for each sample, with velocities and the number of processive events plotted using GraphPad Prism 7, using ANOVA with Tukey's multiple comparison to test significance.

**Online supplemental material**
Fig. S1 illustrates the meiotic LINC complex, dynein, dynactin, and the role of dynein in meiotic prophase I. It also demonstrates the effect of IC2 and LIS1 depletion on dynein and dynactin recruitment to KASH5. Fig. S2 shows immunoblot analysis demonstrating the depletion of targets by RNAi. Fig. S3 shows that LICs 1 and 2 act redundantly in endocytic organelle positioning, with helix 1 being essential. Fig. S4 demonstrates that LICs act redundantly to recruit dynein and dynactin to RILP-positive late endosomes and that dynein recruitment requires helix 1. Fig. S5 depicts the effect of BAPTA-AM treatment on dynein recruitment to KASH5 and Rab11-FIP3, and that KASH5–dynein interactions are not affected by EGTA in vitro. Table S1 contains the quantitation of blots from Fig. 1 C and Fig. 3 E. Tables S2, S3, and S4 document the full multinomial logistic regression analysis of the data shown in Fig. 5 C, Fig. 8 E, and Fig. S3, C and D.

**Data availability**
The data underlying all immunofluorescence figures and graphs are openly available in FigShare at DOI:10.48420/22012580. Images of uncropped blots are available as source data figures.

## Acknowledgments
We thank Quentin Roebuck and Peter Stanley for excellent technical assistance. The Bioimaging Facility microscopes used in this study were purchased with grants from BBSRC, Wellcome, and the University of Manchester Strategic Fund. We would like to thank Drs. Peter March and Graham Wright for their help with microscopy in Manchester and Singapore, respectively. We are grateful to Dr. Alessandra Calvi for generating the nesprin2α construct.

The work in V. Allan's laboratory was funded by the Biotechnology and Biological Sciences Research Council (BB/N006933/1 and a Ph.D. studentship to E. Granger), the Medical Research Council (Ph.D. studentship to C. Villemant), and the University of Manchester. A. Salter was funded by the University of Manchester-A*STAR Research Attachment Programme (ARAP). The work in B. Burke's laboratory was funded by the Singapore Biomedical Research Council and the Singapore Agency for Science Technology and Research, A*STAR. O. Davies is funded by a Wellcome Senior Research Fellowship (grant number 219413/Z/19/Z). C. Lau was funded by grants from the Wellcome Trust (WT210711) and the Medical Research Council, UK (MC_UP_A025_1011), awarded to Dr. Andrew Carter.

Author contributions: Conceptualization: V.J. Allan and B. Burke. Funding acquisition: V.J. Allan and B. Burke. Designed, performed experiments, and analyzed data: A. Salter, K.E.L. Garner, V.J. Allan, C.M. Villemant, E.P. Granger, M. Gurusaran, C.K. Lau, and O.R. Davies. Generated molecular reagents: G. McNee, C.M. Villement, A. Salter, V. Allan, E.P. Granger, and K.E.L. Garner. Purified proteins: C.K. Lau and M. Gurusaran. Supervision: V.J. Allan, B. Burke, P.G. Woodman, and O.R. Davies. Visualization/data presentation: A. Salter, K.E.L. Garner, V.J. Allan, C.M. Villemant, E.P. Granger, M. Gurusaran, and C.K. Lau. Writing the initial draft: A. Salter, K. Garner, V.J. Allan, and C.K. Lau. Review & editing: all authors.

Disclosures: The authors declare no competing interests exist.

Submitted: 13 April 2022

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

# Supplemental material

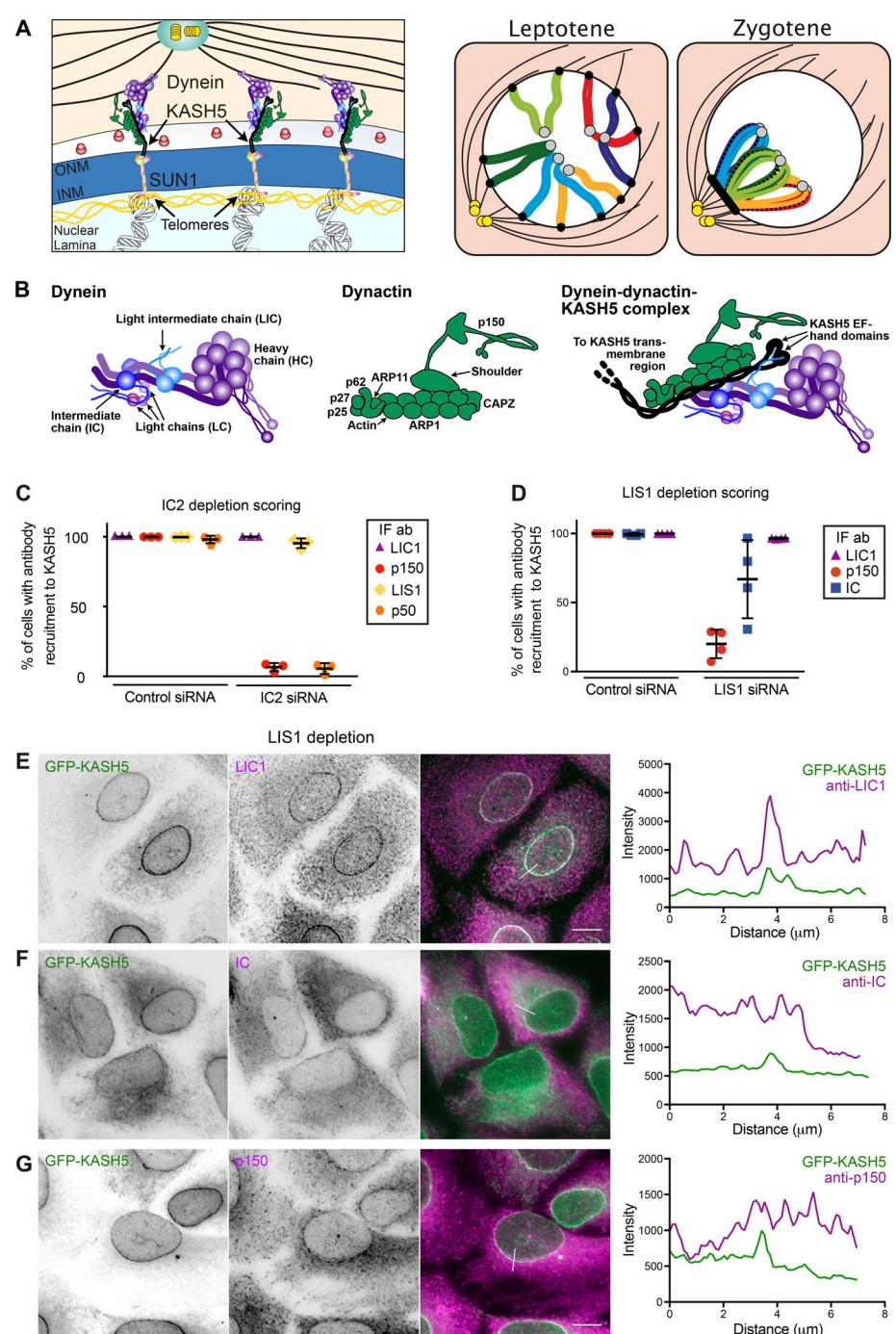

Figure S1. **Effect of IC2 and LIS1 depletion on dynein and dynactin recruitment to KASH5. (A)** Illustration depicting how the meiotic LINC complex connects telomeres inside the nucleus to cytoplasmic dynein in the cytoplasm. The telomeres (black dots) associate with SUN1 or SUN2 at the inner nuclear membrane (INM). KASH5 couples to dynein at the outer nuclear membrane (ONM), dragging the telomeres towards the centrosome (yellow dots) in leptotene of prophase I, generating the chromosome "bouquet" in zygotene. This movement allows the pairing of homologous chromosomes (shades of green, blue, and red/orange) and the formation of the synaptonemal complex (black bars). For simplicity, only six chromosomes are depicted. **(B)** A schematic showing the subunits of dynein, dynactin, and a DDA motor-adaptor complex. **(C)** HeLa cells stably expressing GFP-KASH5 (green) were depleted of IC2 using 20 nM siRNA for 72 h and then processed for immunofluorescence with antibodies against LIC1, dynactin p150, and LIS1. Cells were scored in a binary fashion for recruitment of IC, LIC1, or p150 to KASH5. The mean and standard deviation are shown for three independent repeats in which 300 cells were scored for each condition per experiment. **(D–G)** HeLa cells stably expressing inducible GFP-KASH5 were depleted of LIS1 using 20 nM siRNA, induced to express GFP-KASH5, and then fixed and labeled. **(D)** Cells were scored in a binary fashion to determine if cells showed recruitment IC, LIC1, or p150 to KASH5. The mean and standard deviation are shown for four independent repeats in which 300 cells were scored for each condition per experiment. Statistical tests were determined as not being appropriate for the data in B or C, as there is no variation in some control conditions. Immunofluorescence images showing GFP-KASH5 in green and antibodies against dynein LIC1 (E), IC (F), and dynactin p150 (G) in magenta. Images were taken on a DeltaVision microscope followed by deconvolution. Images are Z-stack projections. Thin white lines on color images are the sites of line scans which are shown on the right. Scale bars represent 10 µm.

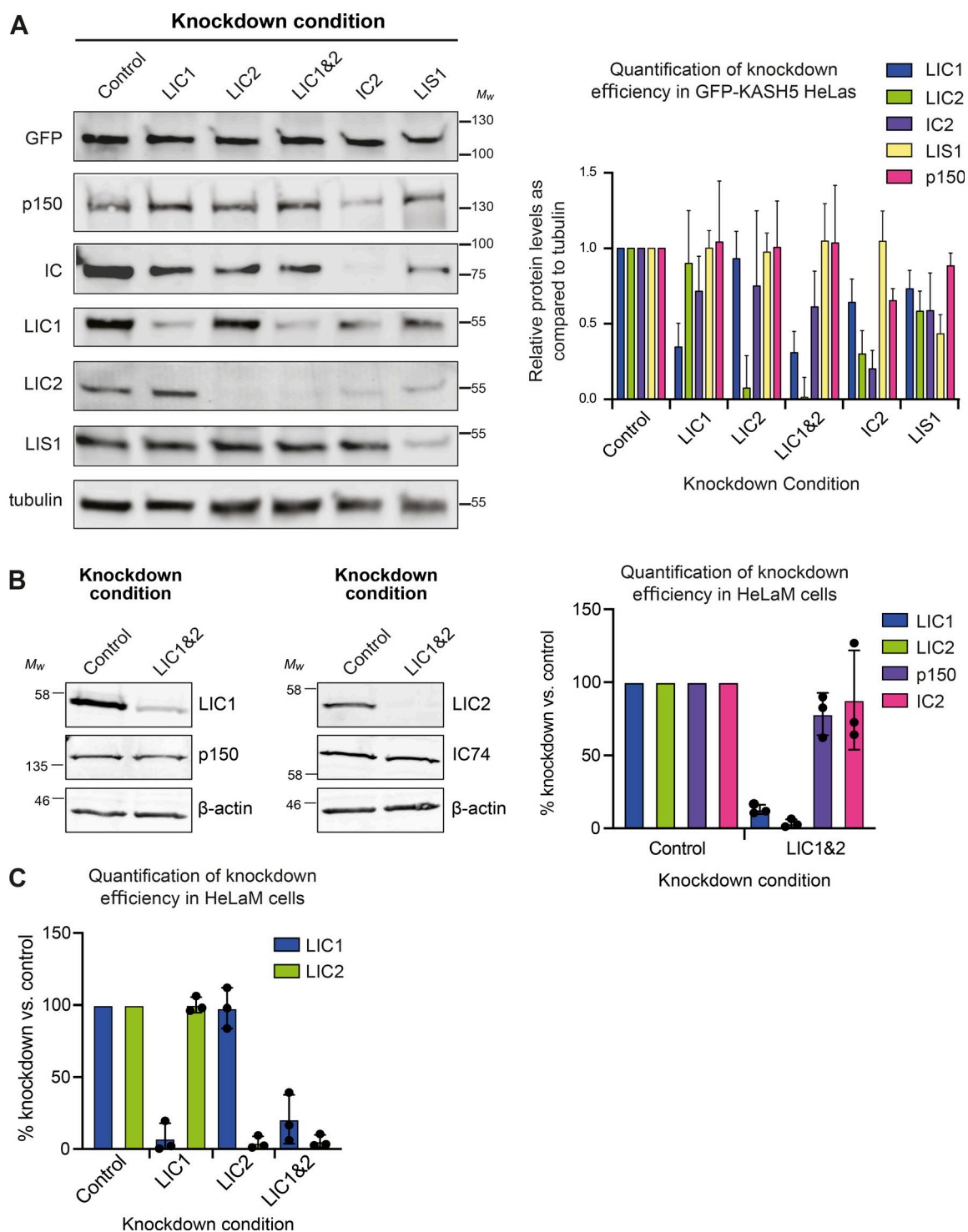

Figure S2. **Immunoblot analysis demonstrating depletion of targets by RNAi. (A)** HeLa cells stably expressing inducible GFP-KASH5 were depleted of various dynein subunits or LIS1 using 20 nM siRNAs against the following targets: Control (Sc), LIC1, LIC2, LIC1&2, IC2, and LIS1. Representative Western blots of cell lysates of knockdown cells are shown. Blots were probed with antibodies against GFP, p150, IC, LIC1, LIC2, LIS1, and α-tubulin. Molecular weight markers are shown on the right. Knockdown efficiency was analyzed by quantification of blots using Image Studio software, with correction for protein loading using the anti-tubulin signal. Experiments were repeated four times alongside immunofluorescence experiments. Error bars represent SD. **(B)** Immunoblotting of HeLaM cells depleted with 5 nM each of LIC1 and LIC2 duplexes, or 10 nM control duplexes, for 72 h. Blots were probed with antibodies to dynein LIC1, LIC2, IC (IC74), and dynactin p150. The mean efficiency of knockdowns (±SD) was analyzed as above, for $n$ = 3 independent experiments. **(C)** Quantitation of siRNA efficiency for LICs individually and together, in HeLaM cells ($n$ = 3, ± SD), including the data depicted in Fig. 5 A. Source data are available for this figure: SourceData FS2.

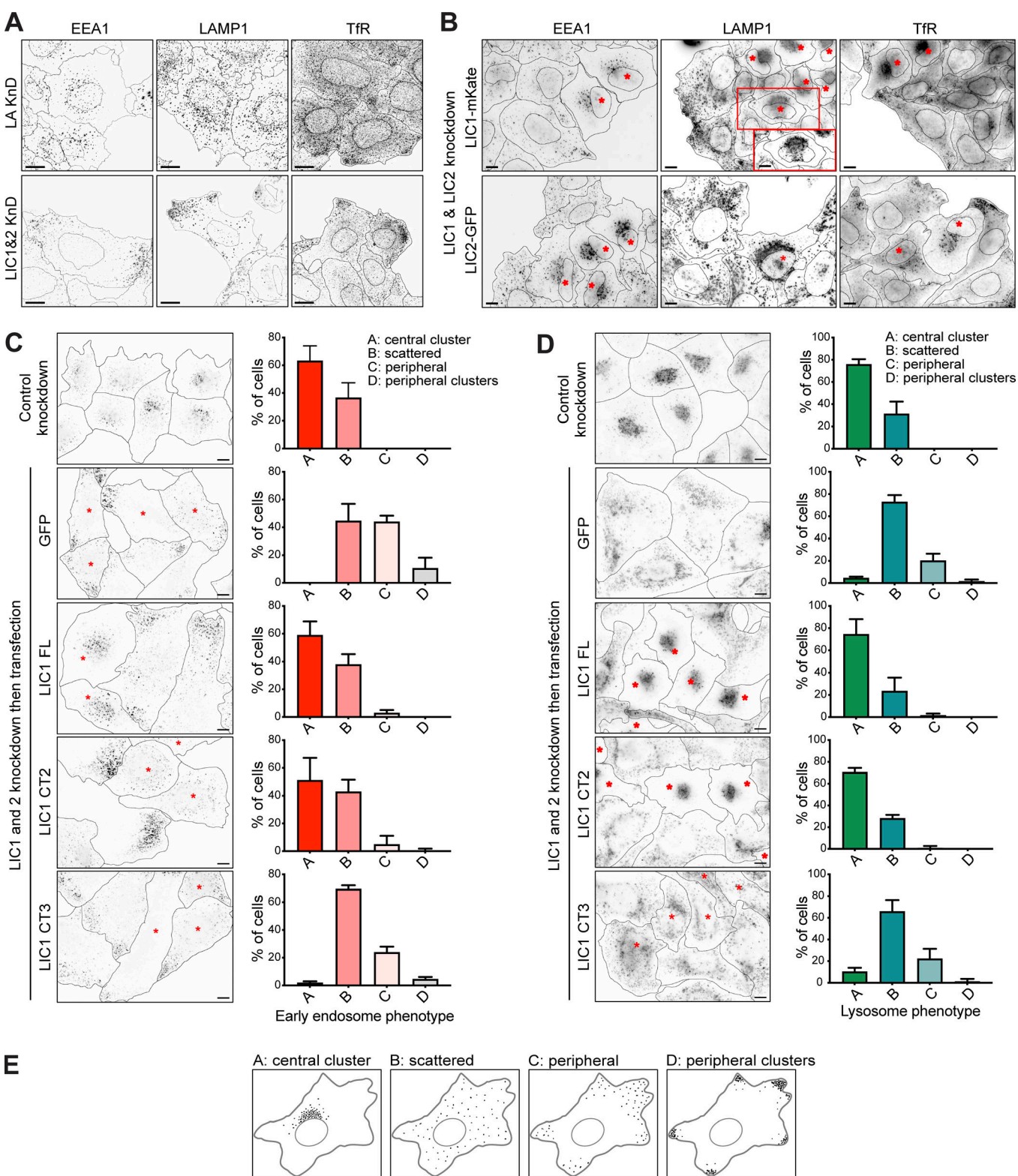

**Figure S3. LICs 1 and 2 act redundantly in endocytic organelle positioning, with helix 1 being essential. (A)** HeLaM cells were depleted of Lamin A/C or LIC 1 and 2 using 5 nM of each siRNA duplex and stained for organelle markers (EEA1, early endosomes; LAMP1, lysosomes; TfR, recycling endosomes). **(B)** LIC-depleted cells were transfected with siRNA-resistant LIC1-mKate or LIC2-GFP and antibody labeled. The boxed region in the LAMP1 image is shown at two different focal planes. Asterisks mark transfected cells. **(C and D)** LIC-depleted cells were transfected with GFP-LIC1 FL, CT2, CT3, or GFP. Asterisks mark transfected cells. Control knockdown cells were not transfected. Cells were labeled with anti-EEA1 (C) or anti-LAMP1 (D). All images are wide-field. Scale bars = 10 µm. **(E)** Early endosome and lysosome position phenotypes were scored as outlined in (E), with ~100 cells per condition, repeated in three independent experiments, with means ± SD plotted. GFP and GFP-LIC1-CT3 rescue data are significantly different to control knockdown (P ≤ 0.0001), whereas rescue with GFP-LIC1-FL or GFP-LIC1-CT2 are not (multinomial logistic regression, see Table S4).

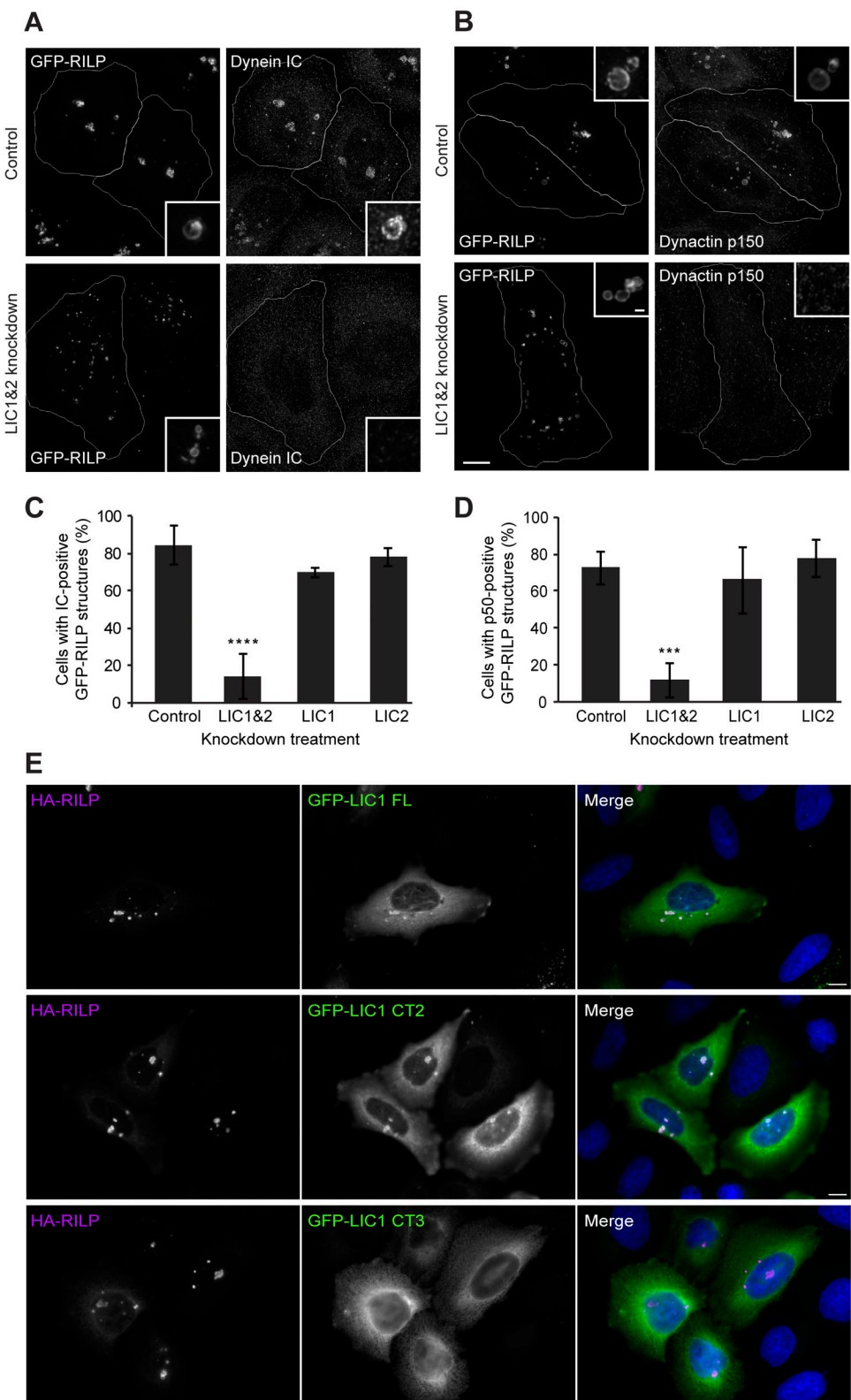

Figure S4.  **LICs act redundantly to recruit dynein and dynactin to RILP-positive late endosomes, and recruitment requires helix 1. (A–D)** Cells were depleted of LICs individually or together for 48 h using a total of 20 nM siRNA then transfected with GFP-RILP and fixed 1 d later and labeled for IC (A) or dynactin p150 (B). Control cells (not siRNA treated) were transfected with GFP-RILP. DeltaVision deconvolved images are shown as z-stack projections. Scale bars = 10 μm in the main image, 0.1 μm in the inset, applicable to all images. The percentage of cells with GFP-RILP structures labeled with IC (C) or p50 (D) was scored (± SD). At least 100 cells were scored in each of the three independent experiments. *** = P < 0.001, **** = P < 0.0001, one-way ANOVA with Dunnett's post-hoc test. **(E)** HeLaM cells were cotransfected with HA-RILP and GFP-LIC1 full-length, CT2, or CT3. Wide-field images are shown; scale bar = 10 μm.

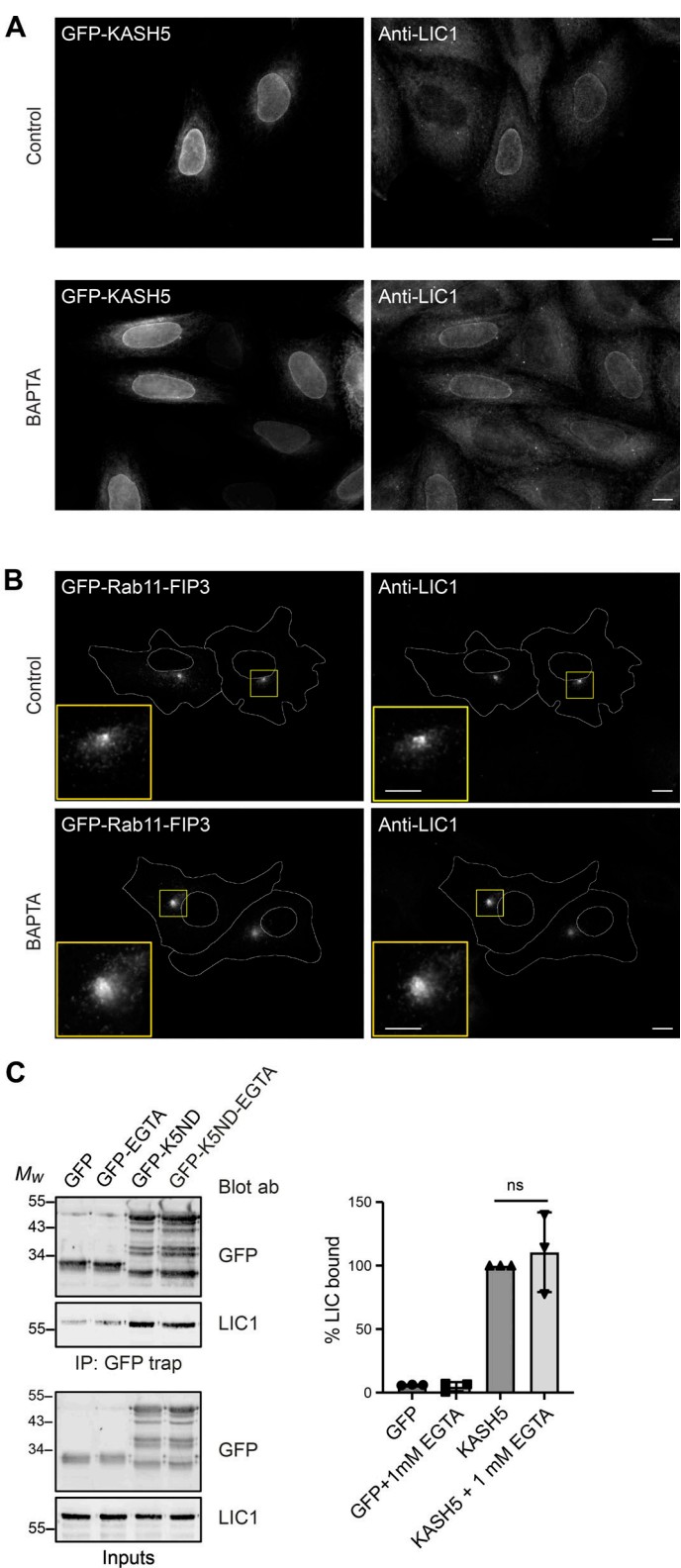

Figure S5. **Effect of calcium depletion on dynein recruitment to KASH5 and Rab11-FIP3. (A and B)** Following transient transfection of HeLa M cells with GFP-KASH5-FL (A) or Vero cells with GFP-Rab11-FIP3 (B), cells were treated with either DMSO vehicle control or 10 μM BAPTA-AM for 2 h at 37°C. Cells were fixed and labeled with antibodies to GFP and endogenous LIC1. Wide-field images, with boxed regions shown as enlargements in B. Scale bars represent 10 μm in main images, 5 μm in insets. **(C)** Lysates of HeLaM cells expressing either GFP or GFP-KASH5ND were used for GFP-trap in the presence or absence of 1 mM EGTA. Beads and inputs (15% of the GFP-trap samples) were probed for endogenous LIC1 or GFP by immunoblotting. An unpaired t test was used to compare levels of LIC1 pulled down with KASH5 ± EGTA (n = 3 independent experiments, P = 0.592). Mean values ± SD are shown.

Provided online are Table S1, Table S2, Table S3, and Table S4. Table S1 shows quantitation of immunoblots shown in Fig 1 C and Fig. 3 E. Table S2 shows multinomial logistic regression statistical analysis of Golgi apparatus morphology following LIC depletions and rescue by GFP and GFP-LIC1 constructs. Table S3 shows multinomial logistic regression statistical analysis of Golgi apparatus morphology following expression of GFP-KASH5ΔK wild-type and EF-hand mutants. Table S4 shows multinomial logistic regression statistical analysis of changes in early endosome and lysosome distribution after LIC depletion and rescue by GFP and GFP-LIC1 constructs.

