## [Peer Review File · The Journal of Cell Biology]

The meiotic LINC complex component KASH5 is an activating adaptor for cytoplasmic dynein

KIRSTEN GARNER, Anna Salter, Clinton Lau, Manickam Gurusaran, Cécile Villemant, Elizabeth Granger, Gavin McNee, Philip Woodman, Owen Davies, Brian Burke, and Victoria Allan

Corresponding Author(s): Victoria Allan, University of Manchester and Brian Burke, ASTAR Skin Research Labs

Review Timeline:

Submission Date:	2022-04-13
Editorial Decision:	2022-05-27
Revision Received:	2022-12-15
Editorial Decision:	2023-02-01
Revision Received:	2023-02-06

Monitoring Editor: Ulrike Kutay

Scientific Editor: Tim Fessenden

Transaction Report:

DOI: <https://doi.org/10.1083/jcb.202204042>

May 27, 2022

Re: JCB manuscript #202204042

Prof. Victoria J Allan
University of Manchester
Faculty of Life Sciences The Michael Smith Building Oxford Road
Manchester M13 9PT
United Kingdom

Dear Prof. Allan,

Thank you for submitting your manuscript entitled "The meiotic LINC complex component KASH5 is an activating adaptor for cytoplasmic dynein". The manuscript was assessed by expert reviewers, whose comments are appended to this letter. We invite you to submit a revision if you can address the reviewers' key concerns, as outlined here.

As you will see, reviewers agreed that the demonstration of a membrane-tethered dynein adaptor for chromosome motility is an important conceptual advance, and that the data presented provide robust support for the main conclusions set forth. Two reviewers sought clarification on data presented in Figure 4, and also requested a justification or alternatives to the indirect binding assays involving the golgi/endosomes. In addition, Reviewer 2 requested quantification of western blots. Finally, Reviewer 3 requested confirmation of KASH5/dynein interaction in meiotic cells. I agree that these data would be a nice addition but I don't consider this confirmation essential for publication. Similarly, additional experimental data beyond these noted here is left to your discretion although all reviewer comments should be addressed in some form.

GENERAL GUIDELINES:

Text limits: Character count for an Article is < 40,000, not including spaces. Count includes title page, abstract, introduction, results, discussion, and acknowledgments. Count does not include materials and methods, figure legends, references, tables, or supplemental legends.

Figures: Articles may have up to 10 main text figures. Figures must be prepared according to the policies outlined in our Instructions to Authors, under Data Presentation, <https://jcb.rupress.org/site/misc/ifora.xhtml>. All figures in accepted manuscripts will be screened prior to publication.

*****IMPORTANT:** It is JCB policy that if requested, original data images must be made available. Failure to provide original images upon request will result in unavoidable delays in publication. Please ensure that you have access to all original microscopy and blot data images before submitting your revision. ***

Supplemental information: There are strict limits on the allowable amount of supplemental data. Articles may have up to 5 supplemental figures. Up to 10 supplemental videos or flash animations are allowed. A summary of all supplemental material should appear at the end of the Materials and methods section.

Please note that JCB now requires authors to submit Source Data used to generate figures containing gels and Western blots with all revised manuscripts. This Source Data consists of fully uncropped and unprocessed images for each gel/blot displayed in the main and supplemental figures. Since your paper includes cropped gel and/or blot images, please be sure to provide one Source Data file for each figure that contains gels and/or blots along with your revised manuscript files. File names for Source Data figures should be alphanumeric without any spaces or special characters (i.e., SourceDataF#, where F# refers to the associated main figure number or SourceDataFS# for those associated with Supplementary figures). The lanes of the gels/blots should be labeled as they are in the associated figure, the place where cropping was applied should be marked (with a box), and molecular weight/size standards should be labeled wherever possible.

The typical timeframe for revisions is three to four months. While most universities and institutes have reopened labs and

allowed researchers to begin working at nearly pre-pandemic levels, we at JCB realize that the lingering effects of the COVID-19 pandemic may still be impacting some aspects of your work, including the acquisition of equipment and reagents. Therefore, if you anticipate any difficulties in meeting this aforementioned revision time limit, please contact us and we can work with you to find an appropriate time frame for resubmission. Please note that papers are generally considered through only one revision cycle, so any revised manuscript will likely be either accepted or rejected.

Thank you for this interesting contribution to Journal of Cell Biology. You can contact us at the journal office with any questions, cellbio@rockefeller.edu or call (212) 327-8588.

Sincerely,

Ulrike Kutay
Monitoring Editor
Journal of Cell Biology

Tim Fessenden
Scientific Editor
Journal of Cell Biology

Reviewer #1 (Comments to the Authors (Required)):

This is a very careful, thorough, and well-done study that elucidates the mechanisms for how the cytoplasmic domain of KASH5 acts as a direct adaptor for cytoplasmic dynein. Given KASH5 importance for fertility, this is a significant paper. It beautifully combines cellular, biochemical, and in vitro assays to convincingly show that the EF hands of KASH5 directly interact with dynein LIC helix 1 to activate dynein/dynactin activity. An interesting part of this is that KASH5 is a membrane protein, and its directly interacting and activating dynein. They then showed by competition assays that it works through dynein recruitment. Overexpression of KASH5 could disrupt other adaptor functions at Golgi and lysosomes. They then did a careful mutant analysis of the EF hand of KASH5 to suggest its not through Ca^{++} . And, they identified the likely mechanism of the zebrafish fue mutant. I also liked that they had single molecule dynein activation assays to see on microtubules how KASH5 activates dynein. It is a complete story and presents a convincing mechanism. It is written very well, and I have very few specific comments. This paper should be of interest to a wide breadth of Cell Biologists, especially those interested in dynein, the nuclear envelope, or organelle trafficking.

Minor comments:

1. Line 69. There aren't 50-60 residues in the lumen. For KASH5 its more like 22 or so. It could be pointed out that KASH5 luminal domain is actually shorter than other Nesprins, missing the Cys at -23.
2. Line 299. In Figure 4B it looks like the LIC CT2 construct might be better recruited than the full length control. Is this important?
3. Line 487. One could argue ZYG-12 was the first trans-membrane activator of dynein. This paper shows it for KASH5 much better, but "first" probably doesn't need to be said.
4. The labels on the x-axis of Fig S1B-C are quite confusing and took me a while to figure out.

Reviewer #2 (Comments to the Authors (Required)):

The study by Garner et al. proposes that the LINC component KASH5 acts as an activating adapter protein for the dynein complex. This is a highly attractive model and would certainly add a lot to our understanding how the LINC complex transmits forces to chromosomes during meiotic chromosome movements. The molecular details how the dynein complex interacts with the KASH protein to transduce force to chromosomes during meiotic chromosome movements have not been elucidated in detail.

Garner et al. define the EF region in the KASH5 protein and the helix 1 domain in dynein light intermediate (LIC) chain as essential. Furthermore, the authors determined the hierarchy of assembly with showing that KASH5 can recruit to dynein light

intermediate chain (LIC) without dynein and that LIS1 is required for functionality of the complex. The dynein intermediate chain (IC) is not needed for dynein and LIS1 recruitment, but for KASH5 association with dynein.

The authors used several different assays to put forward their model. Even though the experimental set-up is somewhat artificial (overexpression of meiotic proteins in HeLa cells) and sometimes extremely indirect (behavior of the golgi in response to KASH5 overexpression), the conclusions are congruent and certainly best supported by the experiments showing that KASH5 supports dynein motility in vitro.

Overall, the paper is carefully composed and clearly written. I am enthusiastic about the paper and would like the following points to be addressed by the authors:

Intro: Better describe the overall effects of meiotic chromosome movements in general: which features are common to all organisms.

Describe in more detail the phenotypes when movement is missing in the model organism of this study.

Are transfection and expression uniform? For the IF images shown in figures 1 ff it is not always entirely clear how often can such a cell be seen. I missed the quantifications. Are these frequently seen examples? How efficient was the transfection with the nesprin-2 α 2 or the mutated KASH5?

I missed the quantifications of the W-blot.

Why is there so much more expression in the helix 2 and helix 1 and 2 LIC truncated constructs?

The authors examined the importance of the KASH5 EF hand domains by using the non-nuclear envelope associated KASH5 mutant proteins. Why was this rather indirect assay used? Why is direct association to KASH5 at the nuclear envelope not shown? The authors make a conclusion based on "a negative result" -the lack of golgi fragmentation-how robust is the fragmentation?

I would recommend to also examine direct association unless there is a reason not to use this assay.

The authors made LIC1 truncations and examined recruitment to the nuclear envelope: the authors showed that in helix 1 and 2 truncated LIC 1 recruitment fails, whereas in the helix 2 deletion recruitment seems to work. Based on these observations the authors conclude that helix 1 is the region required. I believe this interpretation is not exactly correct. It was not ruled out what the effect of deleting helix 1 alone would be. Formally, there could still be cooperation between the domains encompassing helix 1 and 2. I would recommend to make the construct LIC with the helix 1 deletion only and subject this to the same assays as were used for the double helix deletion. If there is a reason to not make this mutant, I would recommend to explain this and tone down in the writing (on several spots in the paper).

The interpretation that LIC1 helix 1 and 2 deletion is specific to interphase it quite far-fetched-it is solely based on the lack of disturbing the mitotic spindle with overexpression of mutant unanchored KASH5 in contrast to disturbing the spindle by a dynein component overexpression. I would suggest to tone down this conclusion in writing.

The model in Supplemental Figure 1 does not fit the used model organism. In mouse, are there not both ends of the chromosome in contact with the nuclear envelope? Why were 6 chromosomes drawn?

Reviewer #3 (Comments to the Authors (Required)):

Summary: In this study the authors dissect the mechanisms by which a meiotic-specific LINC complex component, KASH5, interacts with dynein and dynein-associated factors in its role as an adapter for the nuclear envelope. As the model system, the authors use heterologous expression of KASH5 in HeLa cells, which allows them to look at the interactions its expression can drive since it is highly divergent from endogenous KASH proteins expressed in HeLa cells. Using this system the authors provide a detailed analysis of the KASH5-dynein/dynactin/LIS1 axis. From a combination of microscopy and biochemistry they come the conclusion that KASH5 interacts with dynein light intermediate chains (LICs) to recruit dynein; the N-terminal EF-hands of KASH5 are also required. Depletion of IC2 was particularly interesting in that this disrupted the recruitment of dynein but not the LICs and altered the distribution of KASH5/LIC1/LIS1 within the nuclear envelope. In a reconstitution system, a KASH-less version of KASH5 stimulated dynein-dynactin-Lis1 motility, albeit less than the canonical activator Hook3. Taken together the authors argue that KASH5 is a novel, membrane-embedded dynein adaptor that can activate its activity.

Overall Assessment: Strengths of the manuscript include a model system useful for dissection of the interactions mediated by KASH5, generally high-quality images and analysis, the largely orthogonal application of biochemistry, the reconstitution of dynein-dynactin-Lis1 motility, and the use of tools that can dissect functions of explicit dynein-dynactin components.

Weaknesses include some statements that are not supported by the data (particularly around the LIC1 constructs in Fig. 4), the inclusion of the rather tangential effects of LIC1 depletion on the Golgi and endosomes, reliance on over-expression of not just KASH5 but also other factors including Sun2, and the lack of functional data demonstrating that the insights gained are relevant

in meiosis. Specific points follow.

Major points:

1. Can the authors comment on/leverage the differences in the distribution of KASH5 (and associated factors) at the nuclear envelope in IC2 siRNA conditions or CC1 over-expression (Fig. 2) compared to control conditions (Fig. 1)? The inhomogeneity is striking and one wonders if it might be related to an imbalance of the dynein/LIS1 being engaged but not dynactin? The authors touch on this around line 356 but do not really leverage the potential.
2. While the authors state that "Full-length GFP-LIC1 was efficiently recruited to KASH5, as expected (Fig. 4 B).", the representative image shown in Fig. 4B is quite underwhelming. Line profiles are also not provided as in Figs. 1-3. As presented, the images do not provide strong evidence that the C-terminal helices play a meaningful role in recruitment because they all appear equivalent, although the biochemistry (Fig. 4C) is more convincing. Particularly given the co-over-expression required to carry out this experiment, greater rigor in exploring this angle is needed if the authors wish to make statements about the contribution of the C-terminal helices to the LIC1 recruitment in the manuscript.
3. Figures 5 and 6, which characterize the role for LIC1 and LIC2 in Golgi and endosome distribution, seems out of place and add little to the overall manuscript. I would suggest removing these figures and the focus on these observations at a minimum to the supplemental materials.
4. The biochemical reconstitution is potentially powerful. The authors refer to these as "preliminary motility assays" and reference several constructs that are not shown. This system appears ideal to address how different domains of KASH5 and/or other components contribute to motility but it is currently under-utilized. At a minimum more study of different KASH5 constructs should be completed.
5. To fully rule out a role for calcium binding by the KASH5 EF hands, a subset of the biochemistry should be repeated with the addition of EGTA rather than only by modulating intracellular calcium.
6. Does the over-expression of KASH5 displace endogenous Nesprins in HeLa cells? Might this also play a role in the effects on endogenous dynein/dynactin components, the Golgi positioning, etc.?
7. Have the authors tested if they would arrive at the same conclusions (broadly) if they co-express Sun1 rather than Sun2? Or does the Sun protein co-expressed playing some role?
8. Given that KASH5 acts during meiosis, can the authors offer any evidence for functional relevance of their findings in meiosis itself? The over-expression of numerous factors required in the HeLa system (KASH5, SUN2, and a dynein-related component) makes this a particularly open question.

Minor points:

1. To be congruent with the literature, the term "inner and outer nuclear membranes" or "nuclear envelope" would be preferable over "inner and outer nuclear envelopes" (line 35);
2. Please write out Sun1 and Sun 2 rather than use the Sun1/2 shorthand (e.g. line 35) until introduced - it is confusing for non-experts.
3. In the Introduction please specify that the description applies to mammalian meiosis. There are many eukaryotic organisms that undergo meiosis but differ from the statements made (e.g. do not have sperm or eggs, do not have synaptonemal complex, use actin not microtubules to move chromosomes, etc.).
4. The authors should clarify this statement: "Inside the nuclear envelope lumen, the 50-60 amino acid C terminal KASH-domain sequence associates with SUN proteins" (line 69-70) as the 50-60 amino acid KASH-domain sequence includes the transmembrane domain, so only a subset of it resides in the lumen.
5. Line 72: The suggestion that Sun1 and Sun2 play redundant roles is confusing given that the Sun1^{-/-} mouse is sterile due to a meiotic defect. As more studies have been done, increasingly there is abundant evidence that Sun1 and Sun2 have many unique functions both in meiosis and in other contexts. At a minimum this blanket statement must be qualified with nuance beyond the three papers cited.
6. Although distantly related and apparently different in the details, it would nonetheless be appropriate to discuss and cite the original description of a KASH protein as an adaptor for dynein - Zyg-12 in worms in the introduction ~line 178.
7. It would be very helpful to the reader to have a cartoon of dynein, dynactin, LIS1 - key components that refer to the genetic perturbations etc. especially to facilitate the introduction of components and the discussion.
8. In setting up the experimental system in Fig. 1 I believe that the point is to build a system in which a KASH5-null cell line (HeLa) is converted into a KASH5-expressing system (by engineering the Dox inducible GFP-KASH5 construct), but if so it is worth emphasizing the concept.
9. Can the authors explain this statement: "Because the stable GFP-KASH5 cell line was resistant to transient transfection, we co-transfected HeLa cells with GFP-KASH5 and myc-SUN2 (to ensure efficient localisation of KASH5 to the nuclear envelope..." - do the authors mean that the constructs could not be delivered to cells or, instead, that high levels of GFP-KASH5 expression led to its mislocalization in the cytoplasm unless SUN2 levels were also increased?
10. It would be interesting to also align the EF hands (Fig. 9A) to more ancient KASH proteins that also play a role in meiosis and act as adaptors for dynein, for example in fungi.
11. Is the mitochondrial adaptor Miro not similar to the KASH5-LIC complex characterized here - a GTPase like domain, EF-hand and a tail anchored membrane protein that interacts with dynein (and kinesins) - this seems an apt parallel.

General comments

We thank the reviewers for their thoughtful and insightful comments on our manuscript. We have addressed their criticisms as outlined in detail below. We have also revised the text to take account of the publication of the KASH5 paper by the DeSantis lab (Agrawal et al., 2022), which was previously mentioned in our manuscript as a preprint.

Reviewer #1

This is a very careful, thorough, and well-done study that elucidates the mechanisms for how the cytoplasmic domain of KASH5 acts as a direct adaptor for cytoplasmic dynein. Given KASH5 importance for fertility, this is a significant paper. It beautifully combines cellular, biochemical, and in vitro assays to convincingly show that the EF hands of KASH5 directly interact with dynein LIC helix 1 to activate dynein/dynactin activity. An interesting part of this is that KASH5 is a membrane protein, and its directly interacting and activating dynein. They then showed by competition assays that it works through dynein recruitment. Overexpression of KASH5 could disrupt other adaptor functions at Golgi and lysosomes. They then did a careful mutant analysis of the EF hand of KASH5 to suggest its not through Ca⁺⁺. And, they identified the likely mechanism of the zebrafish fue mutant. I also liked that they had single molecule dynein activation assays to see on microtubules how KASH5 activates dynein. It is a complete story and presents a convincing mechanism. It is written very well, and I have very few specific comments. This paper should be of interest to a wide breadth of Cell Biologists, especially those interested in dynein, the nuclear envelope, or organelle trafficking.

Minor comments:

1. Line 69. There aren't 50-60 residues in the lumen. For KASH5 its more like 22 or so. It could be pointed out that KASH5 luminal domain is actually shorter than other Nesprins, missing the Cys at -23.

We apologise for this inaccurate description, which has now been corrected.

2. Line 299. In Figure 4B it looks like the LIC CT2 construct might be better recruited than the full length control. Is this important?

The apparent difference in expression is primarily down to cell-to-cell variation seen by IF, but it is also the case that CT2 (helix 2 truncation) expresses a bit better than FL or CT3 (helix 1 and 2 truncation) biochemically. It also pulls down somewhat more efficiently with the N-terminal domain of KASH5 than WT LIC1 (Fig. 4B), but this was not the case for the longer KASH5-ΔK construct. We have now scored the recruitment of CT2 and CT3 to KASH5 at the nuclear envelope in more than 230 cells across multiple experiments (Fig. 4E) and show an image of both WT and CT2 as examples of recruitment, and lack of recruitment, respectively. Furthermore, we have greatly strengthened our conclusion that helix 1 is required for dynein-KASH5 interactions by making the point mutations and helix deletion mutants suggested by reviewer 2.

3. Line 487. One could argue ZYG-12 was the first trans-membrane activator of dynein. This paper shows it for KASH5 much better, but "first" probably doesn't need to be said.

Strictly speaking, ZYG-12 has not been directly shown to be a dynein activator. However, we take the point, and have removed the "first".

4. The labels on the x-axis of Fig S1B-C are quite confusing and took me a while to figure out.

We thank the referee for pointing this out, and have revised the labelling of these graphs to make the meaning clearer.

Reviewer #2

The study by Garner et al. proposes that the LINC component KASH5 acts as an activating adapter protein for the dynein complex. This is a highly attractive model and would certainly add a lot to our understanding how the LINC complex transmits forces to chromosomes during meiotic chromosome movements. The molecular details how the dynein complex interacts with the KASH protein to transduce force to chromosomes during meiotic chromosome movements have not been elucidated in detail.

Garner et al. define the EF region in the KASH5 protein and the helix 1 domain in dynein light intermediate (LIC) chain as essential. Furthermore, the authors determined the hierarchy of assembly with showing that KASH5 can recruit to dynein light intermediate chain (LIC) without dynactin and that LIS1 is required for functionality of the complex. The dynein intermediate chain (IC) is not needed for dynein and LIS1 recruitment, but for KASH5 association with dynactin.

The authors used several different assays to put forward their model. Even though the experimental set-up is somewhat artificial (overexpression of meiotic proteins in HeLa cells) and sometimes extremely indirect (behavior of the Golgi in response to KASH5 overexpression), the conclusions are congruent and certainly best supported by the experiments showing that KASH5 supports dynein motility *in vitro*.

Overall, the paper is carefully composed and clearly written. I am enthusiastic about the paper and would like the following points to be addressed by the authors:

Intro: Better describe the overall effects of meiotic chromosome movements in general: which features are common to all organisms. Describe in more detail the phenotypes when movement is missing in the model organism of this study.

We have revised the introduction to clarify these points. We have also made it clear in the abstract that we are talking about mammalian meiosis.

Are transfection and expression uniform? For the IF images shown in figures 1 ff it is not always entirely clear how often can such a cell be seen. I missed the quantifications. Are these frequently seen examples? How efficient was the transfection with the nesprin-2α2 or the mutated KASH5?

An average of $66 \pm 13.5\%$ (SD, n=4 experiments scored) of cells in the GFP-KASH5 cell line express KASH5 after induction. Of these, a small percentage are over-expressing KASH5 in the ER, and these cells correspond to the number that express KASH5 without induction. These cells were always excluded from any analysis. We did not score the recruitment of antibodies to KASH5 nuclei in untreated cells shown in Fig. 1. However, we scored KASH5-expressing cells treated with control siRNA duplexes for recruitment of dynein, dynactin and LIS1 as part of Fig. 2 (scored in Sup. Fig. 1B, C) and Fig. 3. Each antibody labelling was repeated three times, with 300 cells scored per antibody per repeat, giving 900 cells in total per antibody. Virtually 100% of KASH5-expressing nuclei recruited dynein, dynactin and LIS1 antibodies. For the biochemical experiments, the typical transfection efficiency for cells transfected using JetPEI (as used for Figs 1C, 3E, 5, S3, 6, and 8) is $71.3\% \pm 12.6\%$ (S.D., n=5 experiments scored).

I missed the quantifications of the W-blot.

We apologise for this omission. We have added the quantitation either as panels within the figure (Fig. 1D, Fig. 8B, Fig. S2B, Fig. S5C), in the figure legend (Fig. 4B and C), in figure S2 (for the blots in fig. 5A) or in table S1 (for Figs 1C and 3E).

Why is there so much more expression in the helix 2 and helix 1 and 2 LIC truncated constructs?

As explained in our response to reviewer 1's point 2 (see above), these problematic data have been removed and replaced with data from experiments using the LIC1 helix 1 mutants as requested (for full details see explanation below).

The authors examined the importance of the KASH5 EF hand domains by using the non-nuclear envelope associated KASH5 mutant proteins. Why was this rather indirect assay used? Why is direct association to KASH5 at the nuclear envelope not shown? The authors make a conclusion based on "a negative result" -the lack of golgi fragmentation-how robust is the fragmentation? I would recommend to also examine direct association unless there is a reason not to use this assay.

We have worked hard to develop conditions to test dynein recruitment to KASH5 and its EF hand mutants at the nuclear envelope by immunofluorescence. We have found that co-expression of either SUN1 or SUN2 at higher levels than we previously used leads to clear recruitment of native dynein to WT KASH5. We have therefore been able to test the effects of EF hand mutants, as now shown in Fig. 8C and quantified in 8D. These results mirror that of the biochemical assay: dynein is recruited to WT KASH5 and the mod-1 mutant, but not to the *fue*, AA or Mod-2 mutants. We have now scored our Golgi morphology assay, which generates an obvious and robust change upon KASH5DK expression due to dynein sequestration. This is now included as a graph in Fig. 8E, which again shows the same effects of EF hand mutants. The Golgi scattering IF images themselves have been removed.

The authors made LIC1 truncations and examined recruitment to the nuclear envelope: the authors showed that in helix 1 and 2 truncated LIC 1 recruitment fails, whereas in the helix 2 deletion recruitment seems to work. Based on these observations the authors conclude that helix 1 is the region required. I believe this interpretation is not exactly correct. It was not ruled out what the effect of deleting helix 1 alone would be. Formally, there could still be cooperation between the domains encompassing helix 1 and 2. I would recommend to make the construct LIC with the helix 1 deletion only and subject this to the same assays as were used for the double helix deletion. If there is a reason to not make this mutant, I would recommend to explain this and tone down in the writing (on several spots in the paper).

Thank you for the suggestion. We have generated the deletion suggested (440-455), and also made two more mutants and another deletion that Celestino et al. (2019) found to prevent interaction between purified LICs and adaptors, namely F447A/F448A, L451A/L452A, and a deletion of amino acids 433-458. All of these mutations greatly reduced the binding of GFP-LIC1 to HA-KASH5DK in pull-down assays, as now shown in Fig. 4C. Because we have improved our method for analysing dynein recruitment to KASH5 at the nuclear envelope as described above, we have been able to score whether or not the LIC1 mutations or deletions affect KASH5 binding in cells. This data has been included in Fig. 4, with panel D giving examples of cells scored as + or – for recruitment, and panel E showing the scoring data from 209-361 cells across 5-7 experiments. These IF data match well with the biochemistry. Altogether, these data confirm our original conclusion that helix 1 is key for the LIC-KASH5 interaction.

The interpretation that LIC1 helix 1 and 2 deletion is specific to interphase it quite far-fetched-it is solely based on the lack of disturbing the mitotic spindle with overexpression of mutant unanchored KASH5 in contrast to disturbing the spindle by a dynactin component overexpression. I would suggest to tone down this conclusion in writing.

We have removed supplementary figure 5 and the discussion of KASH5 in mitosis, as we agree that it is too preliminary.

The model in Supplemental Figure 1 does not fit the used model organism. In mouse, are there not both ends of the chromosome in contact with the nuclear envelope? Why were 6 chromosomes drawn?

We apologise for the inaccuracy of the figure, which was over-simplified. We have now re-drawn it to show both ends of the chromosomes associated with the nuclear envelope, and have made the bouquet stage clearer. We have chosen to illustrate 6 pairs of chromosomes for simplicity. This is now explained in the figure legend.

Reviewer #3

Summary: In this study the authors dissect the mechanisms by which a meiotic-specific LINC complex component, KASH5, interacts with dynein and dynein-associated factors in its role as an adapter for the nuclear envelope. As the model system, the authors use heterologous expression of KASH5 in HeLa cells, which allows them to look at the interactions its expression can drive since it is highly divergent from endogenous KASH proteins expressed in HeLa cells. Using this system the authors provide a detailed analysis of the KASH5-dynein/dynactin/LIS1 axis. From a combination of microscopy and biochemistry they come the conclusion that KASH5 interacts with dynein light intermediate chains (LICs) to recruit dynein; the N-terminal EF-hands of KASH5 are also required. Depletion of IC2 was particularly interesting in that this disrupted the recruitment of dynactin but not the LICs and altered the distribution of KASH5/LIC1/LIS1 within the nuclear envelope. In a reconstitution system, a KASH-less version of KASH5 stimulated dynein-dynactin-Lis1 motility, albeit less than the canonical activator Hook3. Taken together the authors argue that KASH5 is a novel, membrane-embedded dynein adaptor that can activate its activity.

Overall Assessment: Strengths of the manuscript include a model system useful for dissection of the interactions mediated by KASH5, generally high-quality images and analysis, the largely orthogonal application of biochemistry, the reconstitution of dynein-dynactin-Lis1 motility, and the use of tools that can dissect functions of explicit dynein-dynactin components. Weaknesses include some statements that are not supported by the data (particularly around the LIC1 constructs in Fig. 4), the inclusion of the rather tangential effects of LIC1 depletion on the Golgi and endosomes, reliance on over-expression of not just KASH5 but also other factors including Sun2, and the lack of functional data demonstrating that the insights gained are relevant in meiosis. Specific points follow.

Major points:

1. Can the authors comment on/leverage the differences in the distribution of KASH5 (and associated factors) at the nuclear envelope in IC2 siRNA conditions or CC1 over-expression (Fig. 2) compared to control conditions (Fig. 1)? The inhomogeneity is striking and one wonders if it might be related to an imbalance of the dynein/LIS1 being engaged but not dynactin? The authors touch on this around line 356 but do not really leverage the potential.

We think that the reviewer is referring to the accumulation of LIC1 signal at one side of the nucleus. This was originally observed in 10 % of control cells expressing KASH5 by Horn et al. We therefore do not think it is specific to IC2 depleted cells or cells over-expressing CC1. We also see an accumulation in foci that suggest the MTOC in non-depleted cells, as mentioned in lines 358-9 in the original manuscript (lines 344-346 in the revised ms) and indicated by arrows and asterisks in Figs. 1 and 2, which is more common than asymmetric distribution.

2. While the authors state that "Full-length GFP-LIC1 was efficiently recruited to KASH5, as expected (Fig. 4 B).", the representative image shown in Fig. 4B is quite underwhelming. Line profiles are also not provided as in Figs. 1-3. As presented, the images do not provide strong evidence that the C-terminal helices play a meaningful role in recruitment because they all appear equivalent, although the biochemistry (Fig. 4C) is more convincing. Particularly given the co-over-expression required to carry out this experiment, greater rigor in exploring this angle is needed if the authors wish to make statements about the contribution of the C-terminal helices to the LIC1 recruitment in the manuscript.

We have completely revised Fig. 4, which now shows biochemical and IF analysis with helix 1 deletions and point mutants, as described in the replies to referees 1 and 2. We believe this important point is now fully supported by our new data.

3. Figures 5 and 6, which characterize the role for LIC1 and LIC2 in Golgi and endosome distribution, seems out of place and add little to the overall manuscript. I would suggest removing these figures and the focus on these observations at a minimum to the supplemental materials.

As the Golgi scattering assay is central to other aspects of the paper, we have kept this as a main figure. However, we have moved the endosome/lysosome figure (previously Fig. 6) to the supplemental material (now Fig. S3) as requested.

4. The biochemical reconstitution is potentially powerful. The authors refer to these as "preliminary motility assays" and reference several constructs that are not shown. This system appears ideal to address how different domains of KASH5 and/or other components contribute to motility but it is currently under-utilized. At a minimum more study of different KASH5 constructs should be completed.

We now include the preliminary kymographs, which show 1-407, 1-470 and 1-570 all activate dynein. We also include data showing that the KASH5 EF-hands, the coiled coils, or a combination of EF-hands and coiled coils alone is not enough to activate dynein/dynactin. In combination with other data in this manuscript these assays show the importance of KASH5's domains in dynein activation.

5. To fully rule out a role for calcium binding by the KASH5 EF hands, a subset of the biochemistry should be repeated with the addition of EGTA rather than only by modulating intracellular calcium.

We have repeated the pull-down assays in the presence and absence of EGTA, as requested, and find this has no effect on the efficiency of binding. This data is presented in Fig. S5.

6. Does the over-expression of KASH5 displace endogenous Nesprins in HeLa cells? Might this also play a role in the effects on endogenous dynein/dynactin components, the Golgi positioning, etc.?

We would like to point out that for all the experiments where we see effects on Golgi positioning, these are performed using a KASH5 deletion that lacks the transmembrane and KASH domain, and so would not affect nesprins.

Most of the dynein/dynactin localisation experiments were performed in the stable, inducible cell line that expresses low levels of KASH5. In these cells, the KASH5 is only seen escaping into the ER in a small proportion of cells, and these were never included in any scoring analysis. We have used antibodies to Nesprin2A (MANNES2A) to label cells expressing GFP-KASH5 at the nuclear envelope, and find no difference in its distribution compared to non-expressing cells.

GFP-KASH5 expression (right) does not affect the distribution of endogenous nesprin 2A (left). Note that the expression level of nesprin 2A is variable between cells, perhaps reflecting cell cycle status.

7. Have the authors tested if they would arrive at the same conclusions (broadly) if they co-express Sun1 rather than Sun2? Or does the Sun protein co-expressed playing some role?

We have compared co-expression of KASH5 with SUN1 or SUN2, and find that both SUNs help retain KASH5 in the nuclear envelope and reduce levels of KASH5 in the ER. Indeed, by expressing higher levels of SUNs than before we are now able to observe clear recruitment

of dynein to KASH5 following transient transfection, which we have made use of in revised figures 4 and 8. We have seen no qualitative difference between SUN1 or SUN2 expression.

8. Given that KASH5 acts during meiosis, can the authors offer any evidence for functional relevance of their findings in meiosis itself? The over-expression of numerous factors required in the HeLa system (KASH5, SUN2, and a dynein-related component) makes this a particularly open question.

We understand that this would be a useful addition for understanding the function of KASH5 in vivo. Indeed, the recent paper by the DeSantis lab (Agrawal et al. 2022, eLife 11:e78201) has used in vivo electroporation into mouse testes to express EF-hand mutants in spermatocytes to look for dominant negative effects. The problem with this approach is that the wild type KASH5 is expressed as normal, and the authors only saw a 27% reduction in p150 levels at telomeres by immunofluorescence with only one of the 3 KASH5 EF hand mutants that had affected the KASH5-dynein interaction in cultured cells. Because of these limitations, and since their results are already published, we have decided not to use this approach. In our opinion, it would not be a justifiable use of animal experimentation.

Minor points:

1. *To be congruent with the literature, the term "inner and outer nuclear membranes" or "nuclear envelope" would be preferable over "inner and outer nuclear envelopes" (line 35);*

Corrected to "nuclear envelope" as requested.

2. *Please write out Sun1 and Sun 2 rather than use the Sun1/2 shorthand (e.g. line 35) until introduced - it is confusing for non-experts.*

Corrected as requested.

3. *In the Introduction please specify that the description applies to mammalian meiosis. There are many eukaryotic organisms that undergo meiosis but differ from the statements made (e.g. do not have sperm or eggs, do not have synaptonemal complex, use actin not microtubules to move chromosomes, etc.).*

We have improved our description of meiosis in the introduction as suggested, and also made it clear in the abstract that we are talking about mammalian meiosis.

4. *The authors should clarify this statement: "Inside the nuclear envelope lumen, the 50-60 amino acid C terminal KASH-domain sequence associates with SUN proteins" (line 69-70) as the 50-60 amino acid KASH-domain sequence includes the transmembrane domain, so only a subset of it resides in the lumen.*

We apologise for this inaccurate description, which has now been corrected.

5. *Line 72: The suggestion that Sun1 and Sun2 play redundant roles is confusing given that the Sun1-/- mouse is sterile due to a meiotic defect. As more studies have been done, increasingly there is abundant evidence that Sun1 and Sun2 have many unique functions both in meiosis and in other contexts. At a minimum this blanket statement must be qualified with nuance beyond the three papers cited.*

Thank you for pointing this out. We have revised this text to make this clearer and added some additional references. We have kept the focus on SUN1 and SUN2 in meiosis, however, to avoid confusion.

6. *Although distantly related and apparently different in the details, it would nonetheless be appropriate to discuss and cite the original description of a KASH protein as an adaptor for dynein - Zyg-12 in worms in the introduction ~line 178.*

We have added a description of ZYG-12 at several points in the introduction.

7. It would be very helpful to the reader to have a cartoon of dynein, dynactin, LIS1 - key components that refer to the genetic perturbations etc. especially to facilitate the introduction of components and the discussion.

We agree that this would be useful. We have provided a schematic in Fig. S1 B showing both dynein and dynactin complexes, and a generic activating adaptor. However, it is still not fully understood how LIS1 interacts with dynein/dynactin, and it would be complicated to illustrate all the possible scenarios. We therefore refer readers to excellent recent reviews.

8. In setting up the experimental system in Fig. 1 I believe that the point is to build a system in which a KASH5-null cell line (HeLa) is converted into a KASH5-expressing system (by engineering the Dox inducible GFP-KASH5 construct), but if so it is worth emphasizing the concept.

Thank you for the suggestion to make this point clearer, which we have done.

9. Can the authors explain this statement: 'Because the stable GFP-KASH5 cell line was resistant to transient transfection, we co-transfected HeLa cells with GFP-KASH5 and myc-SUN2 (to ensure efficient localisation of KASH5 to the nuclear envelope...' - do the authors mean that the constructs could not be delivered to cells or, instead, that high levels of GFP-KASH5 expression led to its mislocalization in the cytoplasm unless SUN2 levels were also increased?

This refers to the bizarre (and annoying) observation that while we can transiently transfect and express other constructs efficiently in non-induced GFP-KASH5 cells, as soon as they are induced by doxycycline addition we see a binary split, with the transiently transfected construct only expressing in the proportion of cells that do not express GFP-KASH5. We have added some text to the results to explain that. We therefore have had to resort to transient co-transfection of KASH5 and other plasmids. We have found that co-expressing SUN1 or SUN2 helps prevent KASH5 escaping into the ER, and also enhanced dynein recruitment. Using stronger SUN expression has transformed our ability to test the effect of KASH5 and LIC1 mutants on KASH5-dynein interactions, as shown in the new data in Figs. 4 and 8.

10. It would be interesting to also align the EF hands (Fig. 9A) to more ancient KASH proteins that also play a role in meiosis and act as adaptors for dynein, for example in fungi.

This is an interesting issue, but we think that adding more comparisons would make this figure (now 8A) too complex.

11. Is the mitochondrial adaptor Miro not similar to the KASH5-LIC complex characterized here - a GTPase like domain, EF-hand and a tail anchored membrane protein that interacts with dynein (and kinesins) - this seems an apt parallel.

The mitochondrial adaptor Miro shares some similar features with KASH5 as the reviewer suggests. However, Miro lacks a long coiled-coil which is considered an essential hallmark of proteins that directly activate dynein. Miro instead associates with the cytosolic adaptors TRAK/Milton, which in turn directly activates dynein similarly to KASH5.

February 1, 2023

RE: JCB Manuscript #202204042R

Prof. Victoria J Allan
University of Manchester
Faculty of Life Sciences The Michael Smith Building Oxford Road
Manchester M13 9PT
United Kingdom

Dear Prof. Allan:

Thank you for submitting your revised manuscript entitled "The meiotic LINC complex component KASH5 is an activating adaptor for cytoplasmic dynein". We would be happy to publish your paper in JCB pending final revisions necessary to meet our formatting guidelines (see details below). We leave any clarifying comments requested by the reviewers to your discretion.

A. MANUSCRIPT ORGANIZATION AND FORMATTING:

Full guidelines are available on our Instructions for Authors page, <http://jcb.rupress.org/submission-guidelines#revised>. Submission of a paper that does not conform to JCB guidelines will delay the acceptance of your manuscript.

1) Text limits: Character count for Articles is < 40,000, not including spaces. Count includes abstract, introduction, results, discussion, and acknowledgments. Count does not include title page, figure legends, materials and methods, references, tables, or supplemental legends.

2) Figures limits: Articles may have up to 10 main figures and 5 supplemental figures/tables.

3) Figure formatting: Scale bars must be present on all microscopy images, including inset magnifications. Molecular weight or nucleic acid size markers must be included on all gel electrophoresis.

** Please add scale bars to all inset images, and add molecular weight labels to Figure 5A.

4) Statistical analysis: Error bars on graphic representations of numerical data must be clearly described in the figure legend. The number of independent data points (n) represented in a graph must be indicated in the legend. Statistical methods should be explained in full in the materials and methods. For figures presenting pooled data the statistical measure should be defined in the figure legends. Please also be sure to indicate the statistical tests used in each of your experiments (either in the figure legend itself or in a separate methods section) as well as the parameters of the test (for example, if you ran a t-test, please indicate if it was one- or two-sided, etc.). Also, if you used parametric tests, please indicate if the data distribution was tested for normality (and if so, how). If not, you must state something to the effect that "Data distribution was assumed to be normal but this was not formally tested."

** Please describe the error bars included in Figure 1D, 4E, Supplementary Figure 3 C/D, and Supplementary Figure 5C.

5) Abstract and title: The abstract should be no longer than 160 words and should communicate the significance of the paper for a general audience. The title should be less than 100 characters including spaces. Make the title concise but accessible to a general readership.

6) Materials and methods: Should be comprehensive and not simply reference a previous publication for details on how an experiment was performed. Please provide full descriptions in the text for readers who may not have access to referenced manuscripts.

7) Please be sure to provide the sequences for all of your primers/oligos and RNAi constructs in the materials and methods. You must also indicate in the methods the source, species, and catalog numbers (where appropriate) for all of your antibodies. Please also indicate the acquisition and quantification methods for immunoblotting/western blots.

8) Microscope image acquisition: The following information must be provided about the acquisition and processing of images:

- a. Make and model of microscope
- b. Type, magnification, and numerical aperture of the objective lenses
- c. Temperature
- d. Imaging medium
- e. Fluorochromes

- f. Camera make and model
 - g. Acquisition software
 - h. Any software used for image processing subsequent to data acquisition. Please include details and types of operations involved (e.g., type of deconvolution, 3D reconstitutions, surface or volume rendering, gamma adjustments, etc.).
- 9) References: There is no limit to the number of references cited in a manuscript. References should be cited parenthetically in the text by author and year of publication. Abbreviate the names of journals according to PubMed.
- 10) Supplemental materials: There are strict limits on the allowable amount of supplemental data. Articles may have up to 5 supplemental figures. Please also note that tables, like figures, should be provided as individual, editable files. A summary of all supplemental material should appear at the end of the Materials and methods section.
- 11) eTOC summary: A ~40-50-word summary that describes the context and significance of the findings for a general readership should be included on the title page. The statement should be written in the present tense and refer to the work in the third person.
- 12) Conflict of interest statement: JCB requires inclusion of a statement in the acknowledgements regarding competing financial interests. If no competing financial interests exist, please include the following statement: "The authors declare no competing financial interests." If competing interests are declared, please follow your statement of these competing interests with the following statement: "The authors declare no further competing financial interests."
- 13) ORCID IDs: ORCID IDs are unique identifiers allowing researchers to create a record of their various scholarly contributions in a single place. At resubmission of your final files, please consider providing an ORCID ID for as many contributing authors as possible.
- 14) A separate author contribution section following the Acknowledgments. All authors should be mentioned and designated by their full names. We encourage use of the CRediT nomenclature.

Please note that JCB now requires authors to submit Source Data used to generate figures containing gels and Western blots with all revised manuscripts. This Source Data consists of fully uncropped and unprocessed images for each gel/blot displayed in the main and supplemental figures. Since your paper includes cropped gel and/or blot images, please be sure to provide one Source Data file for each figure that contains gels and/or blots along with your revised manuscript files. File names for Source Data figures should be alphanumeric without any spaces or special characters (i.e., SourceDataF#, where F# refers to the associated main figure number or SourceDataFS# for those associated with Supplementary figures). The lanes of the gels/blots should be labeled as they are in the associated figure, the place where cropping was applied should be marked (with a box), and molecular weight/size standards should be labeled wherever possible. Source Data files will be made available to reviewers during evaluation of revised manuscripts and, if your paper is eventually published in JCB, the files will be directly linked to specific figures in the published article.

WHEN APPROPRIATE: The source code for all custom computational methods published in JCB must be made freely available as supplemental material hosted at www.jcb.org. Please contact the JCB Editorial Office to find out how to submit your custom macros, code for custom algorithms, etc. Generally, these are provided as raw code in a .txt file or as other file types in a .zip file. Please also include a one-sentence summary of each file in the Online Supplemental Material paragraph of your manuscript.

Journal of Cell Biology now requires a data availability statement for all research article submissions. These statements will be published in the article directly above the Acknowledgments. The statement should address all data underlying the research presented in the manuscript. Please visit the JCB instructions for authors for guidelines and examples of statements at (<https://rupress.org/jcb/pages/editorial-policies#data-availability-statement>).

B. FINAL FILES:

-- Cover images: If you have any striking images related to this story, we would be happy to consider them for inclusion on the

journal cover. Submitted images may also be chosen for highlighting on the journal table of contents or JCB homepage carousel. Images should be uploaded as TIFF or EPS files and must be at least 300 dpi resolution.

****It is JCB policy that if requested, original data images must be made available to the editors. Failure to provide original images upon request will result in unavoidable delays in publication. Please ensure that you have access to all original data images prior to final submission.****

****The license to publish form must be signed before your manuscript can be sent to production. A link to the electronic license to publish form will be sent to the corresponding author only. Please take a moment to check your funder requirements before choosing the appropriate license.****

Thank you for this interesting contribution, we look forward to publishing your paper in Journal of Cell Biology.

Sincerely,

Ulrike Kutay
Monitoring Editor
Journal of Cell Biology

Tim Fessenden
Scientific Editor
Journal of Cell Biology

Reviewer #2 (Comments to the Authors (Required)):

The submitted revision is thorough and addressed all points that I raised. This manuscript highlights a very important finding.

Reviewer #3 (Comments to the Authors (Required)):

As this is a revision, my comments here primarily reflect the responsiveness of the reviewers to the prior critiques as stated by the reviewers of the initial manuscript. As the reviewers were quite positive about the impact of the study and its quality, here I address 1) if their requests for additional analyses or further editing to ensure that the data support the authors' claims have been addressed and 2) any issues I found with clarity or the ability to interpret the experiments that could be readily addressed.

1) Overall I found the authors to be highly responsive in this revision, having added additional experimental evidence to support key claims or to have removed claims that could not be unambiguously made.

2) For clarity, I would ask the authors to consider:

- Providing the reader with a bit more information on the prior knowledge about the complement of SUN and KASH proteins expressed in the germline to enable a framework for considering the HeLa system (which nicely shows specificity about KASH5 activity). This is particularly relevant for considering why SUN2 might need to be over-expressed for example, or to consider how the observations conveyed in the manuscript impinge on chromosome movement in germ cells. In other words, the experimental system is defined by adding KASH5 to HeLa LINC complexes, but what other differences are we aware of between these two systems? For example, should we consider the germline-specific SUN proteins?

- In order to set the expectation for how a disruption of recruitment by mutants of KASH5 will manifest (or the influence of knock-down, for example of LIC1&2 in Fig. 3C), it seems that comparisons with HeLa cells not expressing GFP-KASH5 are essential.

Perhaps I simply failed to locate this information, but ideally this would be shown as a comparison throughout.

- Particularly as the authors use the trick of boosting SUN2 expression it is important to cite and discuss evidence that the balance of SUN1 and SUN2 impacts Golgi morphology in the absence of KASH5 but in a manner related to the influence of KASH5 expression. For example, see Hieda et al., Scientific Reports, 2021.